# Virtual epilepsy patient cohort: Generation and evaluation

**Borana Dollomaja**[ID][1*], **Huifang E. Wang**[ID][1*], **Maxime Guye**[2,3], **Julia Makhalova**[2,3,4], **Fabrice Bartolomei**[1,4], **Viktor K. Jirsa**[ID][1*]

**1** Institut de Neurosciences des Systèmes (INS) UMR1106, INSERM, Aix-Marseille Université, Marseille, France, **2** CRMBM, CNRS, Aix-Marseille Université, Marseille, France, **3** CEMEREM, Timone University Hospital, APHM, Marseille, France, **4** Epileptology and Clinical Neurophysiology Department, Timone Hospital, APHM, Marseille, France

* borana.dollomaja@univ-amu.fr (BD); huyfang.wang@univ-amu.fr (HEW); viktor.jirsa@univ-amu.fr (VKJ)

**Data availability statement:** The virtual epilepsy patient cohort has been uploaded to

## Abstract

Epilepsy is a prevalent brain disorder, characterized by sudden, abnormal brain activity, making it difficult to live with. One-third of people with epilepsy do not respond to anti-epileptic drugs. Drug-resistant epilepsy is treated with brain surgery. Successful surgical treatment relies on identifying brain regions responsible for seizure onset, known as epileptogenic zones (EZ). Despite various methods for EZ estimation, evaluating their efficacy remains challenging due to a lack of ground truth for empirical data. To address this, we generated and evaluated a cohort of 30 virtual epilepsy patients, using patient-specific anatomical and functional data from 30 real drug-resistant epilepsy patients. This personalized modeling approach, based on each patient's brain data, is called a virtual brain twin. For each virtual patient, we provided data that included anatomically parcellated brain regions, structural connectivity, reconstructed intracranial electrodes, simulated brain activity at both the brain region and electrode levels, and key parameters of the virtual brain twin. These key parameters, which include the EZ hypothesis, serve as the ground truth for simulated brain activity. For each virtual brain twin, we generated synthetic spontaneous seizures, stimulation-induced seizures and interictal activity. We systematically evaluated these simulated brain signals by quantitatively comparing them against their corresponding empirical intracranial recordings. Simulated signals based on patient-specific EZ captured spatio-temporal seizure generation and propagation. Through in-silico exploration of stimulation parameters, we also demonstrated the role of patient-specific stimulation location and amplitude in reproducing empirically stimulated seizures. The virtual epileptic cohort is openly available, and can be used to systematically evaluate methods for the estimation of EZ or source localization using ground truth EZ parameters and source signals.

## Author summary

People with drug-resistant epilepsy suffer from intractable seizures and can be treated with brain surgery. Their chance of becoming seizure-free heavily relies on proper iden-

the European Brain Research Infrastructure (Ebrains). The dataset card can be accessed via the following link: https://doi.org/10.25493/XWG2-S8X. The code used to generate the synthetic data and perform the analyses presented in this manuscript is available at: https://github.com/BalanceKey/virtual_epilepsy_patient_cohort.git. The cortical surface was parcellated according to the VEP atlas (code available at https://github.com/HuifangWang/VEP_atlas_shared.git).

**Funding:** VKJ acknowledges the support of EU's Horizon Europe Programme under the Specific Grant Agreement No. 101147319 (EBRAINS 2.0 Project), Specific Grant Agreement No. 101137289 (Virtual Brain Twin Project), the support of Agence Nationale de la Recherche under France 2030, reference No. ANR-24-RRII-0005 (NAUTILUS). HEW acknowledges the support of Amidex Recherche Blanc, No. AMX-22-RE AB-135 (HR-VEP Project). The funders had no role in study design, data collection and analysis, decision to publish, or preparation of the manuscript.

**Competing interests:** The authors have declared that no competing interests exist.

tification of brain regions responsible for seizure generation, known as epileptogenic zones (EZ). In the clinic, empirical recordings of brain activity are used to estimate these brain regions. However, the underlying ground truth is not available. We built a virtual cohort of 30 patients with drug-resistant epilepsy, using individual patients' data to simulate their epileptic brain activity. This brain modeling approach based on personalized data is referred to as a virtual brain twin. The key parameters of these virtual brain twins including the EZ settings serve as the ground truth. Epileptic brain activity is simulated at the whole-brain level and mapped onto intracranial sensors to mimic real recordings. We quantified how well the synthetic signals captured features from the empirical data. For each virtual brain twin, we generated spontaneous seizures, stimulated seizures and interictal activity. This cohort is available to the scientific community to benchmark methods such as estimation of EZ and source localization in epilepsy.

## Introduction

Epilepsy is one of the most common neurological disorders, affecting 1% of the global population. It is characterized by recurrent spontaneous seizures, which are sudden bursts of abnormal electrical activity of the brain. Anti-epileptic drugs are the most common treatment option; however one-third of patients with epilepsy are drug-resistant. In such cases, brain surgery is the main alternative treatment, which seeks to resect brain zones responsible for seizures, known as epileptogenic zones (EZ) [1,2].

For precise localization of the EZ before surgery, intracranial depth electrodes (stereoelectroencephalography, SEEG), are inserted in the patient's brain to locally record electrical brain activity [3]. Spontaneous seizures recorded with intracranial electrodes are used to define the EZ. However, due to partial sampling from the intracranial electrodes, spontaneous seizures may not be sufficient to make a clear diagnosis. Intracranial electrical stimulation is performed to contribute to the EZ localization by triggering seizures and for functionality mapping of regions explored [4]. Outside of seizure events and when a patient is at rest, interictal activity is recorded, where interictal spikes are analyzed to contribute to the EZ localization. Both ictal and interical activities are used to constrain and define the EZ.

Despite huge research and clinical efforts to tackle drug-resistant epilepsy, brain surgery has a failure rate of about 50% [1]. Treatment failure is attributed to a misdiagnosis or incomplete diagnosis of the EZ. Therefore, a precise EZ diagnosis is crucial to improve treatment of drug-resistant epilepsy. Many methods have been proposed to diagnose the EZ based on analysis of empirical brain recordings [5,6]. However, they are difficult to evaluate due to absence of ground truth information for empirical data. Recent studies evoke the need for synthetic datasets in order to benchmark scientific methods [7,8]. When synthetic datasets capture the structure and features of empirical data, they can be useful for hypothesis testing and validation before accessing the real dataset [7]. Furthermore, synthetic health care datasets protect patient privacy and are easier to access compared to empirical data for which strict privacy laws are in place [9,10]. Synthetic SEEG data have been used to validate dynamical network biomarkers [11] and validate methods for estimating epileptogenic zones [12,13].

We aimed to build a reliable synthetic dataset of patients with drug-resistant epilepsy, which mimics key features of the empirical data. This resource could be used by the scientific community to test and validate their EZ diagnosis methods. To achieve this, we built for each patient a virtual brain twin, using their own anatomical and functional data [14]. T1-weighted magnetic resonance imaging (T1-MRI) and diffusion-weighted MRI (DW-MRI) defined the structural scaffold with network nodes (brain regions) and links (structural connectivity).

SEEG electrode locations are reconstructed from post operative CT. To model brain activity at each node, the Epileptor model is employed, which describes spatio-temporal seizure dynamics [15]. Finally, we used the Virtual Epileptic Patient (VEP) pipeline [13] to determine the key parameters related to EZ from empirical spontaneous seizure recordings, using personalized whole-brain modeling and machine learning methods [12]. This pipeline is currently used in a clinical trial, with the goal of improving surgical outcome for DRE (EPINOV NCT03643016) [13,16]. As a second approach, we used the clinical EZ estimation from expert epileptologists.

As a result, we generated a virtual cohort of 30 drug-resistant epilepsy patients. To account for synthetic seizures triggered by SEEG stimulation, we extended the Epileptor model. Each virtual brain twin contains synthetic spontaneous seizures, stimulated seizures and interictal activity at the whole-brain level and at the SEEG level. The novelty of our work is the generation of a comprehensive virtual patient cohort, enabling researchers to evaluate their methods using synthetic data that mimic empirical data. Each patient's estimated EZ serves as model parameters, providing its known ground truth from the modeling setting and detailed parameters. We introduced metrics to systematically compare this virtual cohort against the corresponding empirical cohort. Additionally, we interrogated the influence of stimulation location and stimulation amplitude on seizure networks. We shared this dataset publicly in iEEG-BIDS format [17] to help the scientific community when systematically evaluating and refining their EZ estimation methods.

The paper begins by outlining the workflow for building personalized brain models for drug-resistant epilepsy. It then provides an overview of the cohort, examples of simulated time series, and a systematic comparison between simulated and empirical SEEG data. We also assess the significance of key model parameters, focusing on the EZ hypothesis and stimulation parameters. Finally, the discussion addresses limitations and future applications for synthetic data in epilepsy.

## Results

### Workflow of the virtual epileptic cohort

The workflow of the virtual epileptic cohort in Fig 1 illustrates the process of generating patient-specific synthetic data using a personalized whole-brain network model derived from patient-specific brain imaging data, and conducting a systematic comparison.

First, we used patient-specific T1-MRI alongside the VEP atlas [18] to parcellate the brain into anatomo-functional relevant regions, represented as point-like sources. The DW-MRI was used to derive the region-to-region connectivity, by counting white matter streamlines to and from each brain region. Secondly, regional brain activity was simulated using the Epileptor model [15]. Based on the EZ hypothesis, we parametrized the model's excitability for each region. We employed two EZ hypotheses: the VEP hypothesis and a clinically defined hypothesis. The VEP hypothesis is estimated from spontaneous seizures using Bayesian inference methods [13]. The VEP hypothesis was evaluated retrospectively by comparing the inferred EZ with the clinical hypothesis and surgical resection. For all patients, the VEP accurately reproduced the clinically defined EZ network with a mean precision of 0.6, showing significant overlap. Despite some discrepancies, the physical distance between their epileptogenic regions was small (mean, 5.67 mm) [13]. Additionally, the VEP was compared to the resected brain regions of 25 patients who underwent surgery. Here, the false discovery rate (FDR) was employed, to measure the proportion of falsely identified EZ regions (i.e. which were not resected) over all identified regions. The VEP hypothesis showed lower FDR in seizure-free

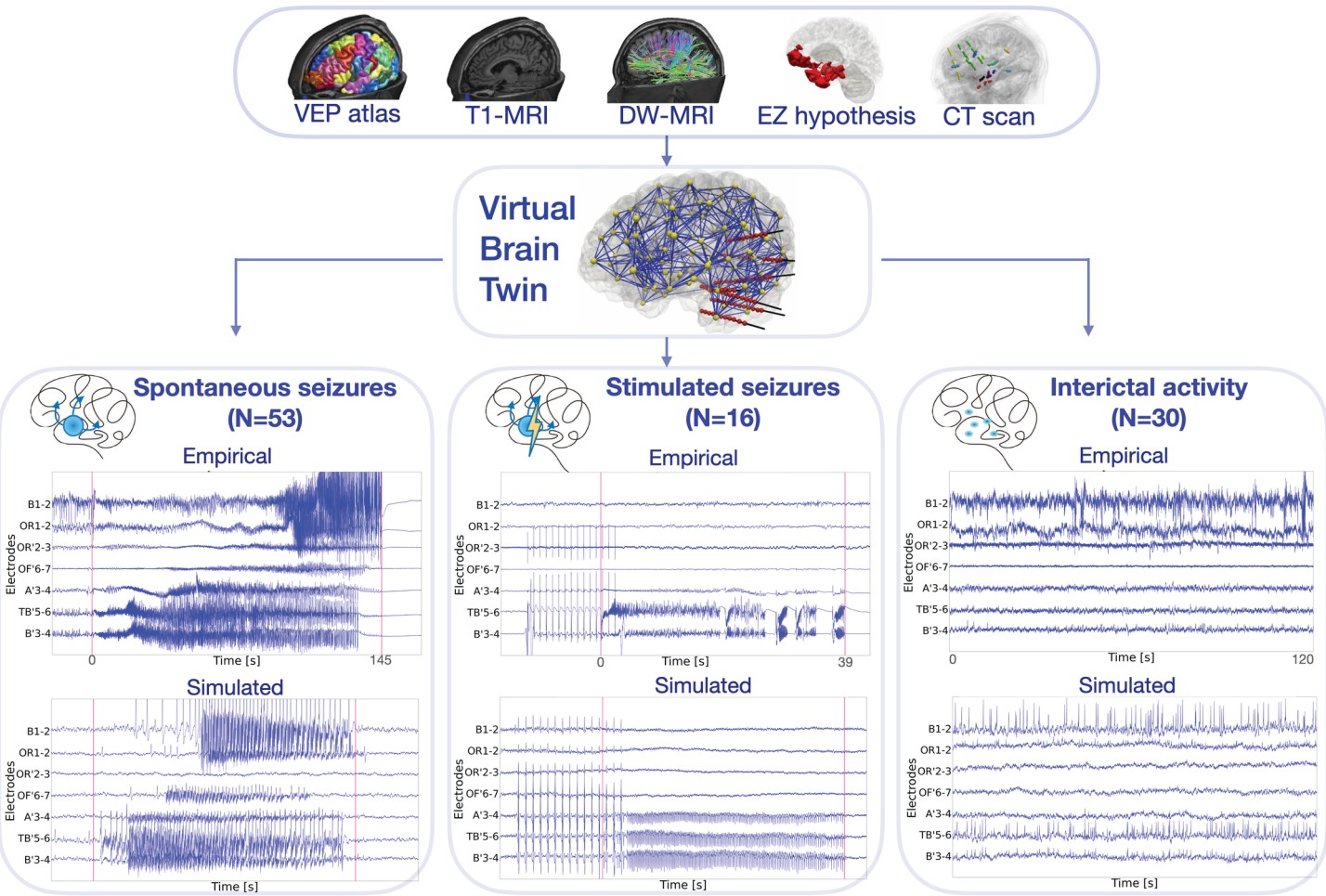

**Fig 1. Workflow of the virtual epileptic cohort**. Patient-specific T1-weighted MRI (T1-MRI), diffusion-weighted MRI (DW-MRI) and CT scan are integrated within a common virtual brain model. The VEP atlas is applied on the T1-MRI to parcellate the brain into anatomical regions, represented as network nodes. From the DW-MRI a tractography is computed, made of estimated white matter fiber connections, from which region-to-region connectivity is derived, represented as network edges. On each node of the virtual brain a neural mass model is used to simulate brain activity. The EZ hypothesis informs the excitability parameter for each brain region and can be derived from ictal SEEG data. Simulated data shown here were generated using the EZ hypothesis from the Virtual Epileptic Patient (VEP) pipeline. A post-SEEG implantation CT scan is used to derive exact electrode locations in the brain. Based on this virtual brain twin, three different states can be simulated: spontaneous seizures, stimulated seizures and interictal spikes. Whole-brain network activity is mapped onto the reconstructed SEEG electrodes using the source-to-sensor forward solution, thus obtaining simulated SEEG time series. The simulated SEEG time series are systematically compared against the empirical SEEG recordings using spatio-temporal data features.

patients (mean, 0.028) compared to those who were not seizure-free (mean, 0.407). This suggests that the VEP could improve predictive power, especially in non-seizure-free cases. The clinical hypothesis is defined by clinical experts (JM and FB). Finally, from post-implantation CT scan we estimated the coordinates of SEEG electrodes. The distance between brain sources and SEEG sensors and the size of source regions is used to evaluate the source-to-sensor gain matrix. The gain matrix maps simulated source-level activity to sensors, obtaining synthetic SEEG activity. For each patient, one personalized brain model was used to simulate three different states: spontaneous seizures, stimulated seizures and the interictal period with spikes (Fig 1).

## Overview of virtual epileptic cohort

The virtual epileptic cohort consists of 30 virtualized drug-resistant epilepsy patients (Table 1) and follows standardized BIDS-IEEG structure [17]. For a detailed overview of this structure, see S1 Fig. Following this standard, the data was categorized into *simulated* data and *derived* data. The simulated data contain simulated brain activity at the SEEG electrode level, with three modalities: spontaneous seizures, stimulated seizures and interictal activity. See Table 1 for a summary of the number of simulations for each modality per patient. The derived data are extracted from MRI and CT-scan brain imaging, capturing the brain anatomy of each patient. This includes: (i) spatial coordinates of brain regions, (ii) coordinates of implanted SEEG electrodes, (iii) region-to-region connectivity matrix, and (iv) source-to-sensor gain matrix. Finally, personalized model parameters (including the EZ hypothesis) and simulations at the whole-brain level are made available. In this paper, we define *ground truth* as the chosen model parameters which gave rise to simulated brain activities at the SEEG level.

**Table 1. Patient information of the virtual epileptic cohort.** Abbreviations: FCD, focal cortical dysplasia; HS, hippocampal sclerosis; L, left; NA, not applicable; PMG, polymicrogyria; PNH, periventricular nodular heterotopia; R, right; *R>L*: right hemisphere onset propagating to left hemisphere; R&L: right and left onset; NSS, number of simulated spontaneous seizures; NIS, number of simulated induced seizures; NII, number of simulated interictal timeseries.

| ID | Sex | Age range | Epilepsy type | MRI | Histopathology | Side | NSS | NIS | NII |
|---|---|---|---|---|---|---|---|---|---|
| 1 | F | 31-35 | Temporal | Normal | HS | R | 2 | 0 | 1 |
| 2 | F | 26-30 | Temporo - occipital | L temporo - occipital PNH | NA | L | 2 | 1 | 1 |
| 3 | M | 36-40 | Temporo - frontal | R temporo - occipital scar | FCD I | R | 1 | 1 | 1 |
| 4 | F | 26-30 | Temporal | R temporal mesial ganglioglioma | Ganglioglioma | R | 2 | 0 | 1 |
| 5 | M | 21-25 | Parietal | L postcentral - parietal gyration asymmetry | NA | L | 2 | 0 | 1 |
| 6 | M | 56-60 | Frontal | Normal | NA | L | 2 | 0 | 1 |
| 7 | M | 56-60 | Temporal | Normal | mild gliosis | *R>L* | 1 | 1 | 1 |
| 8 | F | 46-50 | Temporal | L amygdala enlargement | mild gliosis | L | 3 | 1 | 1 |
| 9 | F | 41-45 | Bifocal: parietal temporal | R parietal lesion | mild gliosis | R | 2 | 1 | 1 |
| 10 | F | 41-45 | Temporal | L hippocampal sclerosis | HS | L | 3 | 1 | 1 |
| 11 | F | 41-45 | Frontal | L frontal scar(abcess) | Gliosis | L | 1 | 1 | 1 |
| 12 | F | 26-30 | Bilateral temporo - frontal | Bilateral hippocampal and amygdala T2-hypersignal | NA | R&L | 2 | 2 | 1 |
| 13 | M | 16-20 | Frontal | Normal | mild gliosis | L | 1 | 0 | 1 |
| 14 | F | 21-25 | Premotor | Normal | FCD IIb | L | 1 | 0 | 1 |
| 15 | M | 41-45 | Temporal | R temporal PMG and multiple PNH | NA | R | 2 | 1 | 1 |
| 16 | M | 26-30 | Temporo - fronto - parietal | R temporo-parieto-insular & L temporo-parietal necrosis | NA | *R>L* | 3 | 0 | 1 |
| 17 | M | 26-30 | Temporal | L temporo-polar hypothrophy and HS | HS | L | 2 | 0 | 1 |
| 18 | M | 21-25 | Parieto - temporal | "L Parieto - occipital necrosis" | NA | L | 1 | 0 | 1 |
| 19 | M | 41-45 | Temporo - insular | Normal | NA | *L>R* | 1 | 0 | 1 |
| 20 | F | 26-30 | Temporal | Normal | HS | R | 1 | 1 | 1 |
| 21 | F | 21-25 | Occipital | Normal | FCD Ic | L | 2 | 0 | 1 |
| 22 | F | 26-30 | Parietal | L parietal FCD | FCD IIb | L | 1 | 1 | 1 |
| 23 | M | 61-65 | Temporal | Normal | NA | L | 1 | 0 | 1 |
| 24 | M | 26-30 | Temporal | Normal | NA | R | 3 | 0 | 1 |
| 25 | M | 41-45 | Insular | Normal | NA | L | 3 | 2 | 1 |
| 26 | F | 26-30 | Occipital | PNH | NA | R | 1 | 0 | 1 |
| 27 | M | 26-30 | Frontal | R prefrontal gliotic scar (arteriovenous malformation) | Gliosis | *R>L* | 1 | 1 | 1 |
| 28 | F | 21-25 | Temporo - frontal | Anterior temporal necrosis | Gliosis | R | 1 | 0 | 1 |
| 29 | F | 26-30 | Bilateral temporal | Bilateral posterior PNH | NA | *R>L* | 3 | 1 | 1 |
| 30 | M | 56-60 | Temporo - frontal | R Frontal FCD | FCD IIb | R | 2 | 0 | 1 |

## Spontaneous seizures

**Simulation of spontaneous seizures.** We simulated spontaneous seizure data for each patient in this virtual epileptic cohort. We used the VEP hypothesis estimated from the VEP pipeline [13]. This pipeline uses patient-specific data to build virtual brain models and Bayesian inference algorithms to obtain estimated distributions of excitability for each brain region, in the [0, 1] range. We linearly transformed epileptogenicity values to the excitability parameter $x_0$ into a range [–2.2,–1.2] in the Epileptor model (3). The critical value $x_{0c}$ is approximately –2.0 depending on the global connectivity matrix [13]. For any given region, if $x_0 > x_{0c}$ the region can autonomously generate seizures, otherwise it remains in the normal state. For each patient, we simulated one seizure for each empirical seizure type. We defined each seizure type qualitatively based on the seizure spatial propagation patterns observed at the SEEG level. While the seizure onset network was the same, the seizure propagation network varied across seizure types (see also S2 Fig). As a second approach, the clinical hypothesis was used to parameterize the epileptogenic network, with results presented in S7 Fig. Epileptogenic, propagation and healthy regions were coupled according to the structural connectome, which gave rise to whole-brain seizure dynamics. We simulated a total of 53 spontaneous seizures for each EZ hypothesis (1–3 per patient).

Examples of spontaneous seizures from two patients of the virtual epileptic cohort (VEC) are shown in Fig 2. Empirical and simulated seizure activity are shown for patient 1 with temporal lobe epilepsy (Fig 2A). Empirical recordings indicated an early recruitment in the hippocampus of the right hemisphere. The estimated epileptogenic network using the VEP pipeline included the hippocampus-anterior, hippocampus-posterior and amygdala of the right hemisphere. The propagation network included the right-STS-anterior, the left-supramarginal-anterior, the left-temporal-pole, the left-rhinal-cortex and the left-Heschl-gyrus. From this estimation, we determined the excitability parameters for all brain nodes and simulated the SEEG time series. The SEEG signal power over the entire activity is visualized on the reconstructed electrodes. Secondly, the same data are shown for patient 5 with parietal epilepsy in Fig 2B. Here, the estimated epileptogenic network included the postcentral-gyrus, superior-parietal-lobule, and angular-gyrus of the left hemisphere. The propagation network extended to both right and left hemisphere brain regions.

**Evaluating spontaneous seizures.** To evaluate the spatio-temporal information captured by the virtual epileptic cohort, we compared simulated and empirical SEEG time series for 53 spontaneous seizures (Fig 3). In addition, to evaluate the importance of personalized parameters in the subsequent simulated data, we built a randomized cohort (RC) of 15 simulated spontaneous seizures. This cohort was constructed using virtual brain models from the VEC cohort and changing only their EZ hypothesis by random selection from another patient.

We showed four metrics to compare empirical and simulated spatio-temporal seizure dynamics in Fig 3 (all 16 metrics shown in S8 Fig). We captured high frequency components of the signal during seizure activity using a high-pass filter followed by envelope smoothing using a low-pass filter. When the envelope crossed a defined threshold, we marked the corresponding electrode as seizure electrode. By marking the timestamps when the envelope jumps from and returns to baseline, we estimated the seizure onset and offset times, respectively (see Fig 3A). Next, each electrode's activity was binarized in time at each timestep (0, no seizure activity; 1, seizure activity) (see Fig 3B). The binarized synthetic and empirical SEEG data were compared using Pearson correlation. We also compared the intersection of SEEG seizure channels between the empirical and synthetic time series, resulting in our overlap metric. Finally, based on seizure onset times, we categorized each channel as either seizure onset

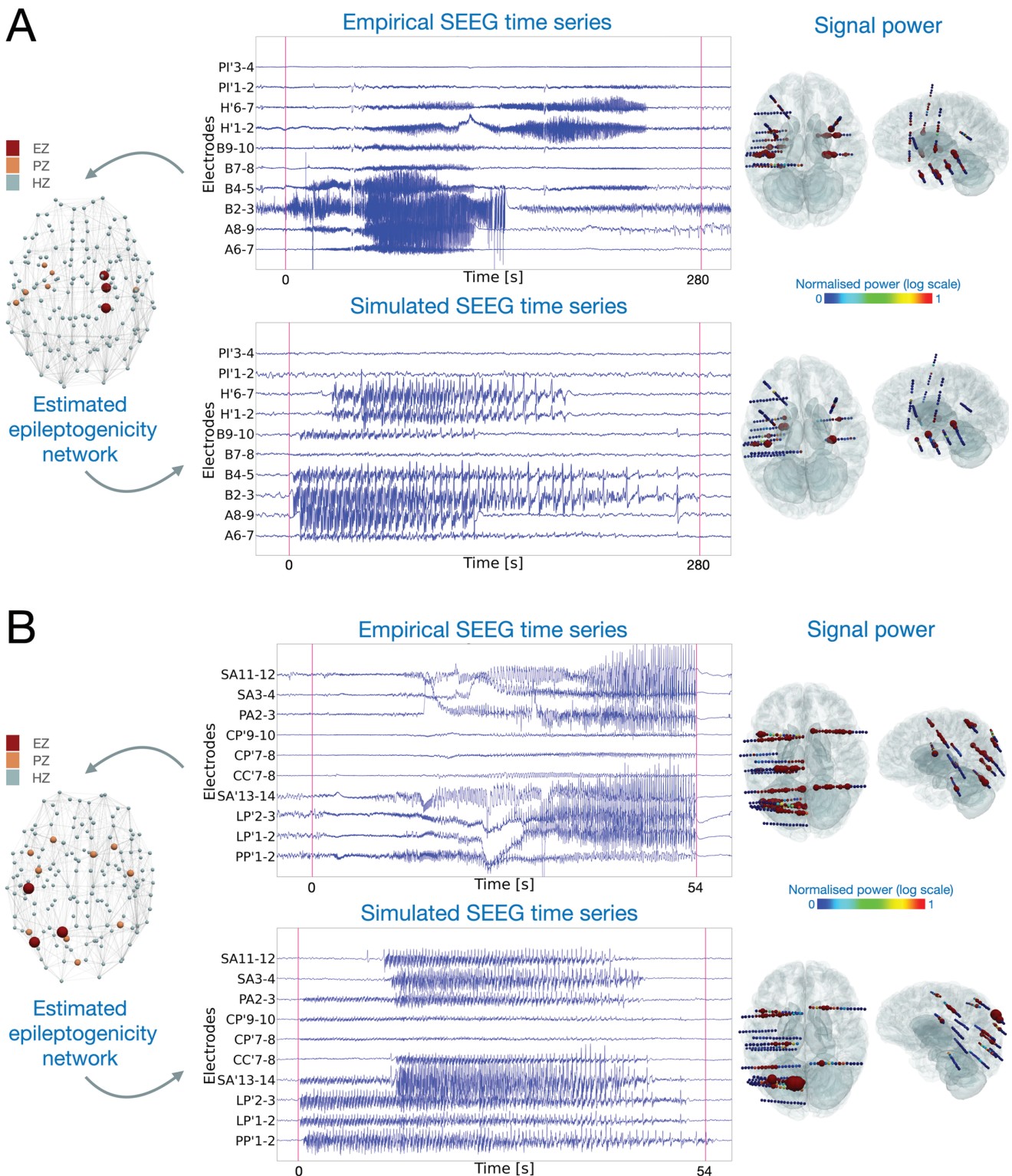

**Fig 2. Corresponding empirical and simulated spontaneous seizure activity.** (**A**) Patient 1 with temporal lobe epilepsy showing right-hemisphere hippocampal early involvement propagating after several seconds in the contra-lateral hemisphere, and (**B**) Patient 5 with parietal epilepsy. In both (A) and (B), on the left-side panel, the estimated epileptogenic network from the empirical spontaneous seizure is plotted for each brain region. Red, orange and light blue represent EZ, PZ and HZ respectively. The top middle panels display empirical SEEG recordings. The bottom middle panels represent corresponding simulated recordings based on the EZ network. Red vertical lines denote the seizure onset and offset, determined by clinicians for the empirical recordings and by our model in the simulated time series. Right-side panels show the signal power distribution for all channels. The colorbar displays normalized signal power, where blue and red represent low and high signal power, respectively.

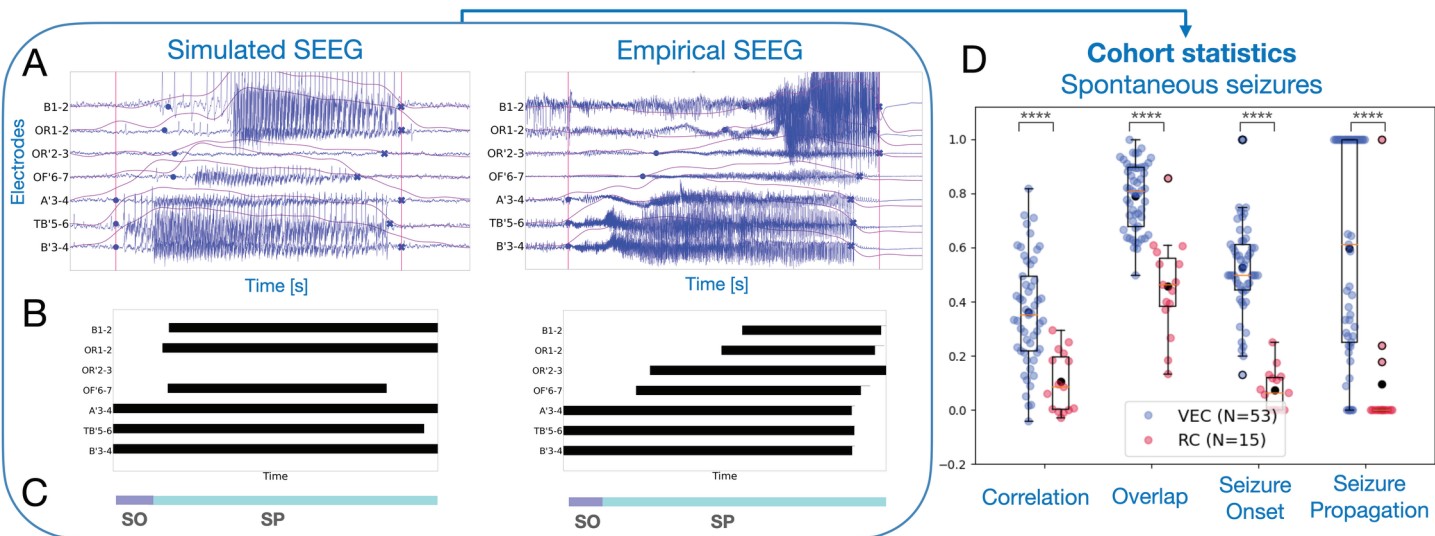

**Fig 3. Comparison of spontaneous simulated SEEG signals with empirical recordings.** (A) Left, simulated SEEG seizure. Right, empirical SEEG timeseries of seizure dynamics. For each electrode, the envelope data feature is computed by band-pass filtering the electrical brain signal. Seizure onset and seizure offset time points, shown as blue circles, are computed for each electrode based on the envelope's jump from and return to baseline. (B) Each empirical and simulated SEEG electrode is binarized in time, where 0 (white) corresponds to no seizure activity and 1 (black) corresponds to seizure activity. (C) The time reference bar is used for both empirical and simulated SEEG to categorize recording electrodes into two groups: SO (Seizure Onset) and SP (Seizure Propagation). An electrode is labeled as SO when its onset time point aligns with the horizontal purple line, typically within the first few seconds of the entire seizure duration. Whereas, an electrode is labeled as SP when its onset time point aligns with the horizontal light blue line. (D) Four metrics to quantify the comparison of SEEG recordings in virtual epilepsy cohort (VEC, $N = 53$ in blue) and the randomized cohort (RC, $N = 15$ in red) against the empirical SEEG recordings. Each point in the swarm plot corresponds to one metric for one empirical and simulated SEEG pair. The black point in each metric represents the mean value for each category. Results are shown in box plots, overlayed over individual data points. Middle box represents the interquartile range (IQR), with a line at the median. The whiskers extend from the box to the data point lying within 1.5× the IQR. Points past the whiskers are marked as fliers. $^{****}p<0.0001$; permutation test.

(SO), seizure propagation (SP) or no seizure. Each category was compared using the Jaccard similarity coefficient (see Fig 3D, see also Materials and methods). We performed a permutation test for each metric ($H_0$ : mean(VEC) ≤ mean(RC), $H_1$ : mean(VEC) > mean(RC)). The virtual epileptic cohort performs significantly better than the randomized cohort ($p<0.0001$). In addition, results with EZ hypothesis based on the clinical hypothesis are presented in S7 FigA.

## Stimulated seizures

**Simulation of stimulated seizures.** We provided 16 SEEG stimulated seizures for 14 patients. We used the EZ hypothesis derived from spontaneous seizures and modeled the electrical stimulus based on the clinical stimulation parameters. Specifically, the spatial distribution of the electric field was computed based on the stimulation location and charge, while the stimulus waveform was modeled according to the stimulation frequency, pulse width, and total duration. Finally, the strength of the electric field was input into the Epileptor-stimulation model, scaled proportionally to the values of its phenomenological variables. The scaling parameter is determined by comparing empirical signals with simulated ones (see Equation 4). In Fig 4A, we show an example of a focal temporal seizure from patient 3, triggered by stimulation of electrodes B2 and B3, as anode and cathode, respectively. The epileptogenic network is estimated from the spontaneous seizure of the same patient using the VEP pipeline. The epileptogenic zones are T2-anterior and hippocampus-anterior of the right hemisphere. The propagation zones are SFS-rostral, amygdala and fusiform gyrus of the

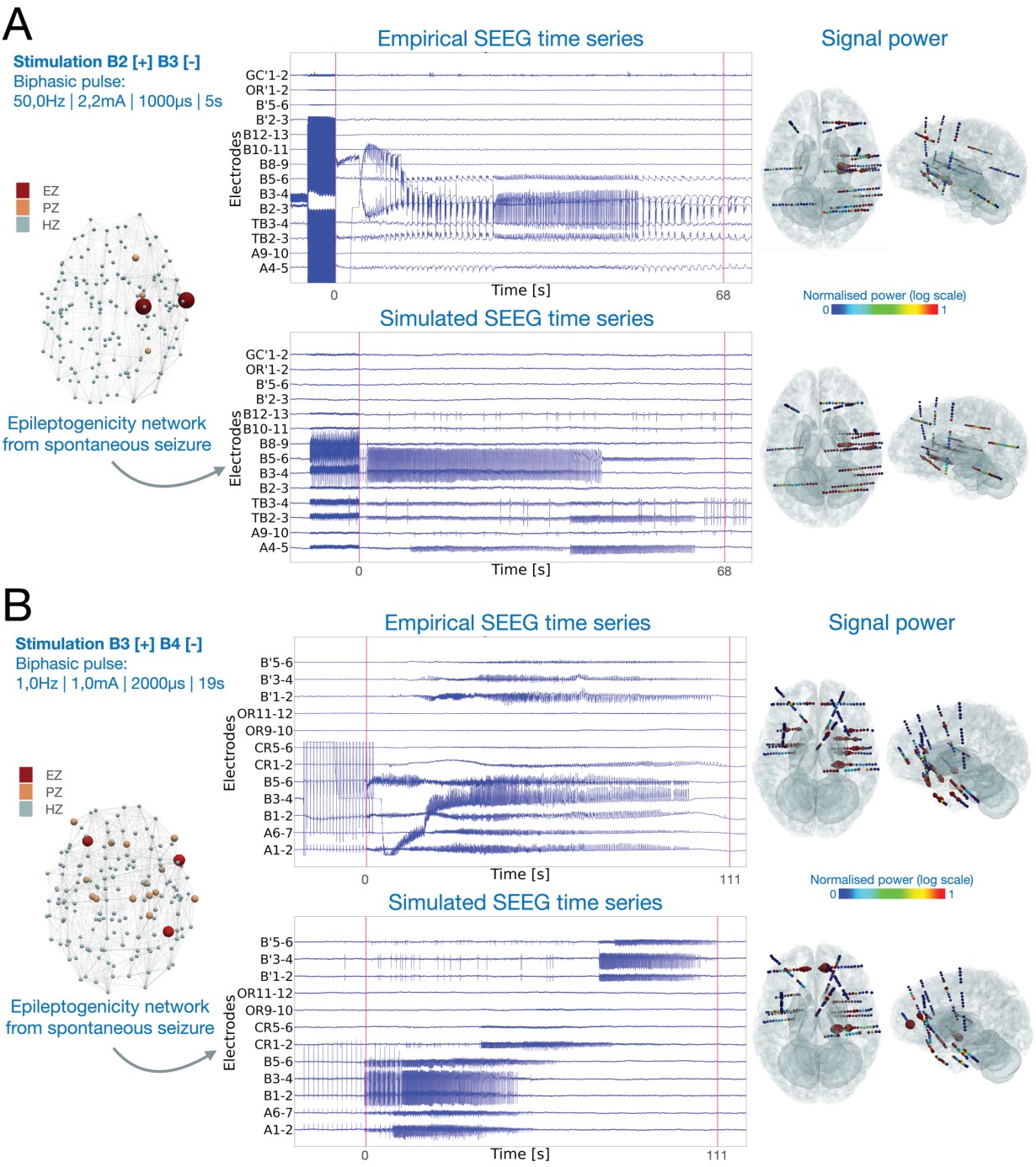

**Fig 4. Stimulated seizures induced by adjacent SEEG electrodes for two patients with different propagation patterns.** (**A**) Focal temporal seizure of patient 3, induced by high-frequency stimulation of electrode pairs (B2–B3), located in the right hippocampus anterior. Stimulation waveform is a bipolar pulse applied at 50 Hz frequency, 2.2 mA amplitude, with pulse width of 1 millisecond and 5 second duration. (**B**) Bilateral seizure of patient 12, induced by low-frequency stimulation of electrode pairs B3-B4, also located in the right hippocampus anterior. Stimulation waveform is a bipolar pulse applied at 1 Hz frequency, 1 mA amplitude, with pulse width of 2 milliseconds and 19 second duration. In both (**A**) and (**B**), left-side panels display stimulation parameters, and the epileptogenic network estimated from the spontaneous seizure of the same patient. Red, orange and light blue represent EZ, PZ and HZ respectively. The middle panels show empirical and simulated SEEG time series for a few electrodes. Red vertical lines denote the seizure onset and offset, determined by clinicians for the empirical recordings and by our model in the simulated time series. The right-side panels show the normalized signal power distribution for all channels. Color bar represents signal power, where blue and red represent low and high signal power, respectively.

right hemisphere. Following the stimulation period, a seizure is triggered in the stimulating electrodes and other nearby electrodes. The second example consists of a bilateral temporo-frontal seizure from patient 12, applied using electrodes B3 and B4 (Fig 4B). The epileptogenic network is estimated from the spontaneous seizure of the same patient. The epileptogenic zones are left orbito-frontal-cortex, right F3-pars-opercularis and right occipito-temporal-sulcus. Propagation zones have an extended network including the right hippocampus-anterior where the stimulating electrodes are located. Seizure activity is first observed in the right-temporal-lobe, and later propagates to frontal regions and the contra lateral hemisphere (B' electrodes located in the left temporal lobe). In both examples, normalized signal power distribution on reconstructed SEEG electrodes displays the large network of seizure organization.

**Evaluating stimulated seizures.** We used three approaches to evaluate synthetic stimulated seizures (Fig 5). Before comparison, we removed the time series corresponding to the stimulus current and only compared the post-stimulus time series. In the first approach, similarly to spontaneous seizures, four metrics were compared against a randomized cohort (all tested metrics in S9 Fig). The randomized cohort was generated using random EZ hypothesis and contains in total 15 stimulated seizures. Thus, for the same patient, an EZ hypothesis was chosen from a random patient. Then, we applied the same stimulation parameters to simulate seizure dynamics induced by stimulation. We employed a permutation test for the comparative metrics that specifically compared means. The results demonstrate a significantly better performance for the virtual epileptic cohort compared to the randomized cohort, as shown in Fig 5A. However, here the seizure propagation metric (SP) showed no significant differences between the two cohorts. Results with EZ hypothesis based on the clinical hypothesis are presented in S7 FigB.

In the two other approaches, we investigated the role of two stimulation parameters in inducing seizures: stimulation location and stimulation amplitude. We varied these stimulation parameters and compared the simulated outcome against the empirical stimulated seizure. We varied stimulation location by randomly selecting 10 electrode pairs from each of four distance groups to stimulate for seven patients, shown in Fig 5B. The distance groups are defined by $d_e$ the distance from the empirical stimulation location: Dist1: $d_e \leq 1$; Dist2: $d_e \in [1, 2]$, Dist3: $d_e \in [2, 3]$ and Dist4: $d_e \geq 3$. For each patient, stimulation parameters are all the same as their empirical cases, except for the stimulation locations. As stimulation location was selected incrementally further away from the empirical location, the similarity between the simulated and empirical seizure dynamics deteriorated, as observed across our four metrics (see also S5 Fig). Both structural connectome and the EZ network configurations determined the stimulated seizure patterns.

For varying stimulation amplitude, we could evaluate the capacity of our model in generating seizure dynamics for a particular stimulation amplitude (Fig 5C). For each patient, first, we adjusted the model parameters to induce seizures by stimulation using the same stimulation amplitude as the empirical case. Then, we varied stimulation amplitude using common amplitudes used clinically (two lower and two higher amplitudes than the empirical one) to simulate the signals and evaluate the post-stimulus response by comparing it with the empirical stimulated seizure. Lower stimulation amplitudes did not induce a seizure, which translated to low similarity values across metrics. Higher stimulation amplitudes induced synthetic seizures which were similar to the empirical amplitudes, but longer lasting (see S4 Fig).

In summary, on average of seven patients, the virtual epileptic cohort is capable of capturing spatio-temporal features of stimulated seizure data based on its personalized EZ hypothesis, stimulation location and stimulation amplitude.

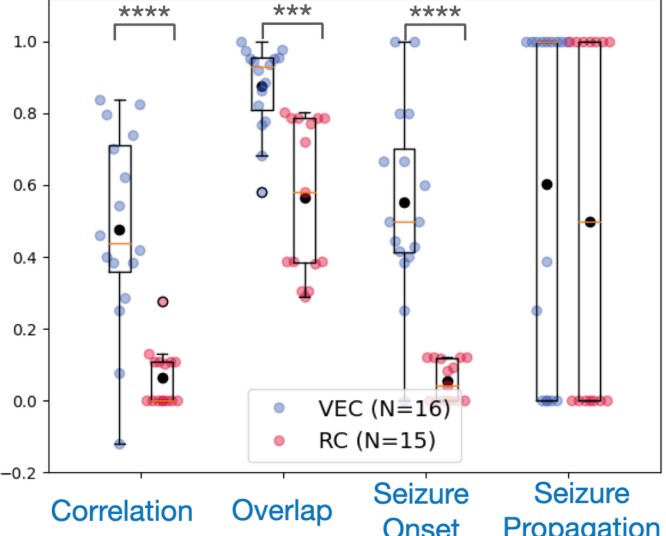

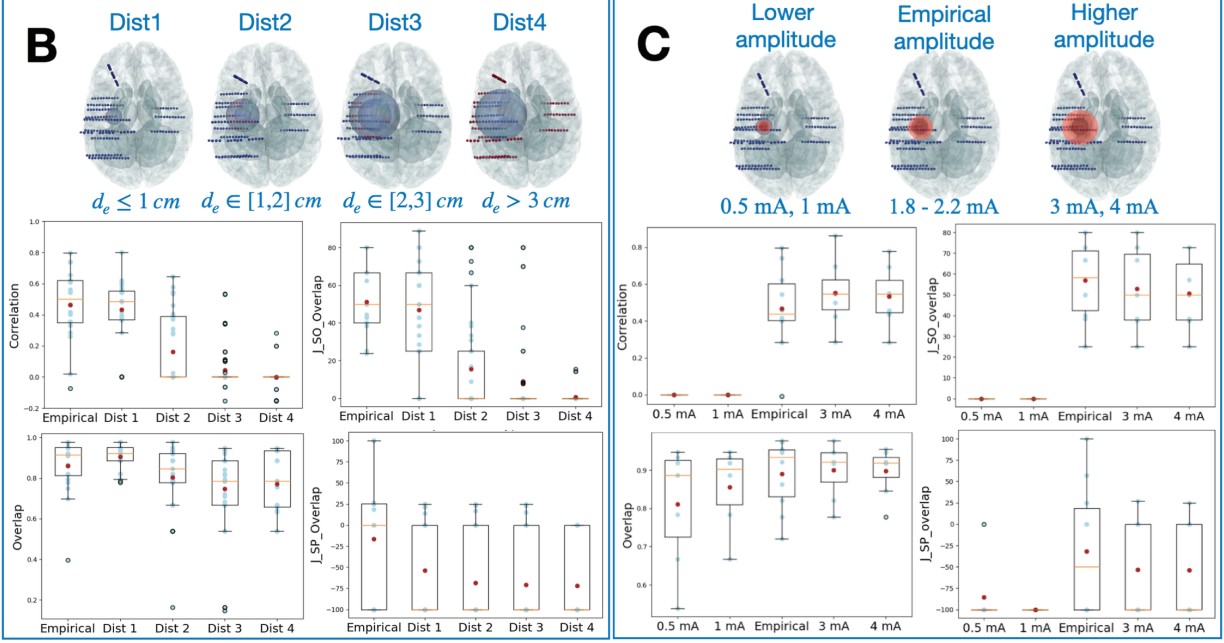

**Fig 5. Comparison of simulated SEEG signals with empirical recordings for stimulation-induced seizures.** (**A**) Four metrics to quantify the comparison of stimulated SEEG seizure time-series in virtual epileptic cohort (VEC, $N = 16$ in blue) and the randomized cohort (RC, $N = 15$ in red). Each point in the swarm plot corresponds to one metric comparing one empirical and simulated SEEG pair. Black points in each metric represent the mean value. $^{****}p<0.0001$, $^{***}p<0.001$; permutation test. (**B**) Performance metrics for varying only stimulation locations in seven patients, measured by the distance from the empirical stimulation locations. We randomly stimulated 10 pairs of electrodes within four main distance groups located outside of the empirical locations. If we define $d_e$ as the distance in cm from the empirical locations, Dist1: $d_e \leq 1$; Dist2: $d_e \in [1, 2]$; Dist3: $d_e \in [2, 3]$; and Dist4: $d_e \geq 3$. Four metrics are used to compare the five distance groups in four box plots for seven patients, with individual data points overlaid (in blue). (**C**) Performance metrics for varying only stimulation amplitude in seven patients. Empirical stimulation amplitude varied from 1.8 to 2.2 mA. We then performed simulations using the same stimulation parameters but 2 lower amplitudes (0.5 mA, 1 mA) and 2 higher amplitudes (3 mA, 4 mA). Four metrics are used to compare across the five stimulation amplitude groups. In all cases, results are shown in box plots, overlaid over individual data points. Middle box represents the interquartile range (IQR), with a line at the median. The whiskers extend from the box to the data point lying within 1.5× the IQR. Points past the whiskers are marked as fliers.

### Interictal spikes

**Simulation of interictal spikes.** In our virtual epileptic cohort, we simulated brain activity and SEEG recordings during the interictal period for all 30 patients. We simulated interictal time series based on the EZ hypothesis, such that the EZ network could generate interictal spiking. We illustrated simulated SEEG and empirical SEEG data for patient 8, shown in Fig 6A. The detailed shapes of interictal spikes in both simulated and empirical SEEG are illustrated as well. We also mapped interictal spike counts across all implanted SEEG electrodes in 3D space (Fig 6B).

**Comparing empirical and simulated interictal spikes.** To evaluate the simulated interictal spikes, we measured interictal spike count for each SEEG electrode, which is the fraction of spikes in each electrode compared to all detected spikes. Spikes were detected after bandpass filtering the signal and identifying peaks above calculated thresholds as described by [19]. Next, we compared the empirical and simulated spike counts across all patients of the virtual epileptic cohort and calculated the correlation of their spike counts in the left box of Fig 6C. In addition, we generated a randomized cohort for 15 patients by using the same brain models but randomly selecting an EZ network from other patients in the cohort. We also calculated spike count correlation with empirical data in the randomized cohort. Our second approach using the clinical hypothesis was also evaluated in S7 FigC. Permutation testing showed a significant difference between the two groups ($H_0$: mean(VEC) ≤ mean(RC); $p<0.001$).

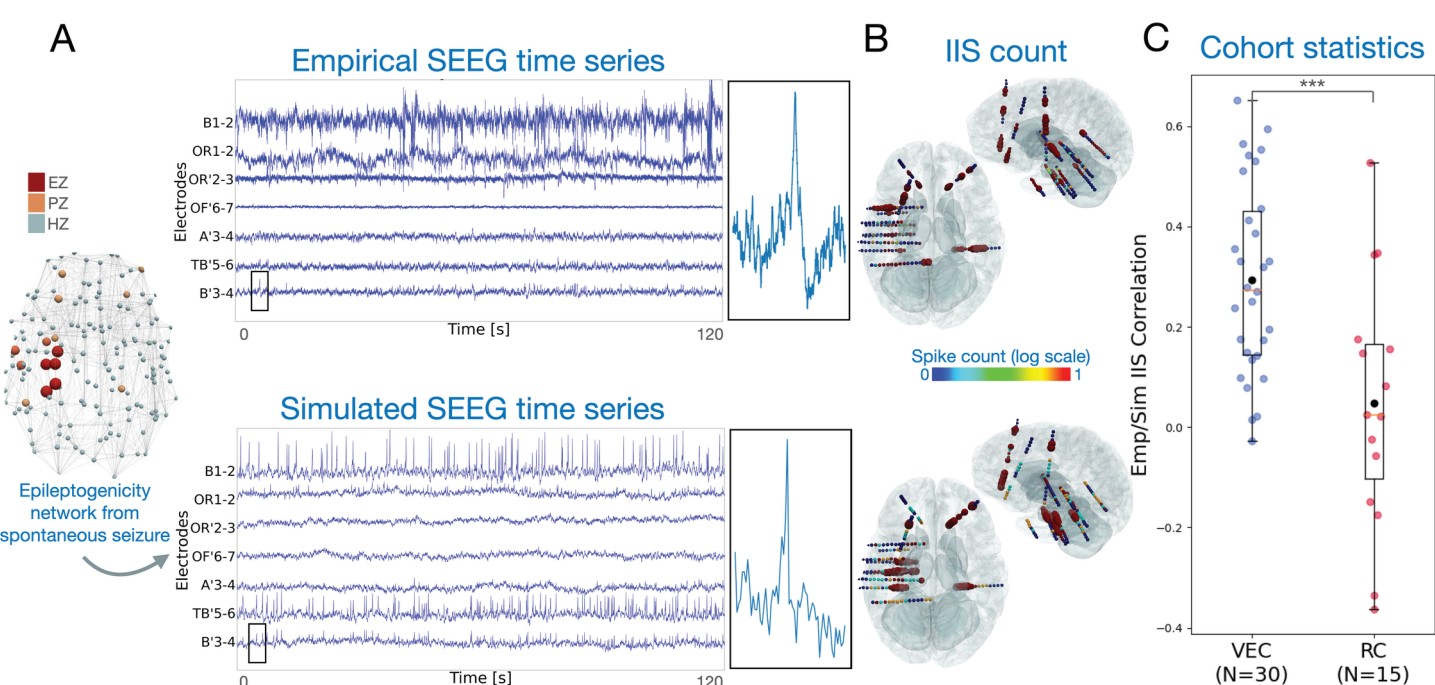

**Fig 6.** (**A**) Interictal activity of one patient shown by SEEG signals in empirical and simulated cases, with interictal spikes in multiple electrodes. One interictal spike is shown by zooming in on channel B'3-4 for both the simulated and empirical cases. (**B**) Interictal spike (IIS) count across all implanted SEEG electrodes are displayed in axial and sagittal view for the simulated and empirical SEEG time series. Spike count values are normalized and displayed in logarithmic scale. (**C**) Boxplots and swarmplots for correlation of simulated and empirical IIS count. Correlation values are plotted for all virtual epileptic cohort patients (VEC, N=30, in blue) and randomized cohort (RC, N=15, in red). Middle box represents the interquartile range (IQR), with a line at the median. The whiskers extend from the box to the data point lying within 1.5x the IQR. Points past the whiskers are marked as fliers. Black points represent the mean value. A permutation test was computed to evaluate the significance of the difference between the two group average values, where [***]$p<0.001$; permutation test.

## Discussion

In this paper, we provided a cohort of 30 virtualized drug-resistant epilepsy patients for hypothesis testing and validation. For each patient, we used one virtual brain model to generate synthetic spontaneous seizures, stimulation-induced seizures and interictal activity with spikes. Here, we presented synthetic stimulated seizures by extending the Epileptor model, which captures spatio-temporal seizure dynamics and interictal spikes [15]. In addition, we systematically evaluated our synthetic SEEG time series by comparing them against their corresponding empirical SEEG recordings. The synthetic SEEG data are simulated using personalized brain models, based on patient-specific brain connectivity, alongside reconstructed brain sources and SEEG sensors [20]. We showed that the synthetic data respect the structure and features of the empirical dataset. In particular, we demonstrated that the parameters used to simulate synthetic SEEG data are important in capturing spatio-temporal features of epileptic activity, such as seizure propagation, stimulation outcome and interictal spike count. These parameters serve as ground truth for evaluating data analysis methods in clinical epilepsy research.

When using the virtual epileptic cohort data, there are several important points to take into account. First, the VEP framework for EZ hypothesis estimation is a proposed methodology undergoing evaluation in a clinical trial, with some preliminary results [13]. It constructs a whole-brain model for a given patient using individual brain imaging and Bayesian inference methods to specify the parameters from seizure data. Second, simulated spontaneous (i.e., unprovoked recurrent) seizures are driven by the slow dynamics of the permittivity variable $z$ in the Epileptor model, which guides the system periodically from normal to seizure-like states. This variable mimics time-dependent changes during seizures as extracellular $[K^+]$, oxygen and ATP consumption [15]. Studies have shown that the transition to seizure is a slow process, involving gradual loss of network resilience [21,22], increased sensitivity to stimulation [23] and ion dynamics changes, including extracellular $[K^+]$ [24], intracellular chloride [25] and extracellular calcium [26], although ionic processes are highly intertwined [27]. However, no universal mechanism is known for seizure generation and several other dynamical models for seizure onset have been proposed [28–35]. Thirdly, we hypothesized that repetitive stimulation causes slow accumulation of ion imbalances which can trigger seizures. To model stimulated (i.e., provoked) seizures, we made $m$ time-varying in the Epileptor model, which influences the stability of the seizure state [36]. During stimulation, $m$ slowly increases until crossing a seizure threshold, pushing the system into the seizure state (see S3 Fig). The seizure threshold was tuned based on the EZ hypothesis and empirical stimulation parameters. Lower stimulation amplitudes failed to cross the threshold, whereas empirical and higher amplitudes succeeded (see S4 Fig). In future studies, this approach can be refined for each patient when more empirical stimulation data are integrated for model inversion. Finally, for both types of simulated seizures, generation of seizure dynamics on a virtual brain depends on the EZ hypothesis and the structural brain connectivity (SC). The EZ hypothesis informs the excitability parameter $x_0$ for each regional Epileptor, while the SC determines the connection weights between them, such that regional brain activity depends on both its excitability and the activity of connected regions [20,37]. This allows synthetic seizures to emerge as a whole-brain phenomenon influenced by local activity and connectivity.

To ensure the quality and usefulness of the synthetic data, we implemented a set of comparative metrics, although to the best of our knowledge there is no standard method for validating synthetic data. We focused on specific data features of the real SEEG signal such as the envelope function, seizure onset and offset times and interictal spikes. In addition, the relationship between seizures, spikes, and stimulated activity is not straightforward. Our VEP

cohort can be a tool for researchers to verify their hypotheses across these three modalities. However, we do not address the full complexity of the epileptic brain in the resting state and background dynamics, which have been shown to contain useful information for delineating the EZ [38]. We focused on simple spatio-temporal network features because, although the Epileptor captures common properties of brain activity during seizures [15], it does not account for all seizure dynamotypes [39]. This was shown by our metrics, where empirical data features were captured by the synthetic data, albeit only to a certain extent. However, the same metrics were applied to a randomized cohort and demonstrated that non-informative parameters fail to capture the same empirical data features. In addition, our simulations were not biased by surgical outcome (S10 Fig).

Furthermore, for stimulated seizures, we systematically varied stimulation location and amplitude and compared the outcome against the empirical data. In the clinical setting, these parameters exhibit the greatest variation. Other parameters such as stimulation frequency are typically chosen as either 1 Hz or 50 Hz, while a standard range is adhered to for pulse width (0.5–3 ms) and stimulation duration (20-60 sec for 1 Hz stimulation; 3–8 s for 50 Hz stimulation) [3]. However, the seizure propagation metric (SP) did not perform as well for stimulated seizures compared to spontaneous seizures, highlighting a limitation of our study. Stimulated seizures in the empirical dataset were marked by expert epileptologists. Additional stimulations which did not trigger seizures or triggered afterdischarges (ADs) were not simulated. ADs are generally not considered to be a reliable predictor of the seizure onset zone and may have different underlying mechanisms from seizures [40,41]. In addition, stimulated seizures resembling a patient's spontaneous seizures, both electrographically and in semiology, are strong predictors of postsurgical outcomes [40]. In our dataset, semiology was not considered, and all stimulated seizures were simulated without distinction. Lastly, we compared spike count correlations between simulated and empirical interictal data. Despite using the same EZ parametrization, the average correlation was lower than that of seizure dynamics. This discrepancy may be explained by studies indicating that propagation zones can generate independent interictal spikes, suggesting the epileptogenic and irritative zone overlap but are not identical [42,43]. The main contribution of these simulated interictal data is their ground truth information that can be used to evaluate data analysis methods for interictal spikes (e.g., source localization).

We simulated spatio-temporal seizure dynamics on 162 brain regions defined in VEP atlas [18]. This low spatial resolution constrains the complexity of simulated propagation patterns, as compared to empirical SEEG data. Each brain source represents on average $\sim$16 cm$^2$ of the cortical surface. Conversely, neural field models use finer spatial scales ($\sim$1 mm$^2$ of the cortical surface) and consider short-range cortical connections in addition to long-range white matter connectivity [44–49]. In addition, dipole orientations of brain sources are not taken into account by the forward solution used to compute the SEEG time series. The current dipole is mainly attributable to pyramidal cells in the cortical gray matter and is aligned perpendicular to the brain surface [50]. Neural field models provide more realistic source to sensor mapping by considering both orientation and distance between the dipole sources and the sensors [48,51]. However, unlike neural field models, neural mass models are typically more efficient in terms of computational resources, simulation duration and parameter exploration. They remain reliable in capturing various features of brain activity, including seizure dynamics [15,51].

The current dataset can be enriched by other data imaging modalities, such as EEG, MEG and fMRI, directly through the forward solution on the provided brain source signals in this cohort. Additional data modalities can also be integrated to better inform the virtual brain twins and improve the personalized synthetic data, such as PET/SPECT [52], sodium MRI

[53], MEG [54] and high-resolution EEG [55]. The models that account for a vast combination of clinical seizure patterns [39], or other biophysically grounded models [56,57] that link parameters closer to underlying biological mechanisms, can also be utilized. Replacing neural mass models by neural field models which account for cortical geometry and short-range connectivity is an additional future goal. This would allow for the integration of high-resolution brain imaging data (e.g. ultra-high field MRI [$\geq 7\,T$]) which can improve model creation and patient specificity [58]. Overall, these informative features and modeling approaches can be integrated in the virtual epileptic cohort to provide a richer repertoire of simulated seizure dynamics.

## Materials and methods

### Ethics statement

The study protocol was approved by the local Ethics Committee (COMITÉ DE PROTEC-TION DES PERSONNES SUD MEDITERRANÉE 1). Informed written consent was obtained for all patients in compliance with ethical requirements of the Declaration of Helsinki.

### Study design

This study consisted of using a cohort of 30 patients with drug-resistant epilepsy alongside a methodology of personalized virtual brain modeling to build a synthetic copy of the dataset, called the virtual epileptic cohort. The objective of this study was to provide researchers and other potential users a virtual dataset with ground truth for testing and validation of their methods. We provided this dataset alongside comparative metrics between the synthetic and empirical data to gauge the capacity of the virtual cohort in capturing relevant data features.

We used non-invasive T1-MRI and DW-MRI to reconstruct patient-specific whole-brain network models. The Epileptor [15] model was used to simulate brain activity for each network node. The Epileptor was extended to account for stimulated seizures, alongside its existing dynamical regimes for interictal and spontaneous ictal dynamics. The model's excitability parameter was defined using the EZ hypothesis, which was estimated using the VEP pipeline [13]. A second EZ hypothesis was defined from a team of clinical experts (JM and FB). For each empirical SEEG recording, a synthetic copy was simulated and was either interictal activity ($N = 30$), spontaneous seizure ($N = 54$) or stimulated seizure ($N = 16$). Simulations were performed on the whole-brain level and were mapped onto reconstructed SEEG electrodes using a source-to-sensor gain matrix. Electrode locations were obtained from cranial CT-scan after electrode implantation. The synthetic SEEG time series were compared against empirical SEEG recordings using 16 metrics comparing spatio-temporal seizure network dynamics and interictal spike organization. We selected four metrics to describe the main data features that were captured, but provided all of them in S8 Fig and S9 Fig. To assess the extent to which a personalized EZ hypothesis captured features of the empirical recordings, we constructed a surrogate cohort of 15 patients using random EZ hypothesis. The randomized cohort was also compared against the empirical recordings. Then, the virtual epileptic cohort and the randomized cohort metrics were compared and a permutation test was applied for significance testing. Finally, stimulation location and amplitude were systematically varied in-silico and compared against empirical stimulated seizures.

### Patient data

**Empirical patient data.** A total of 30 retrospective patients with drug-resistant epilepsy underwent a standard presurgical protocol at La Timone hospital in Marseille, France. All

patients underwent comprehensive presurgical assessment, including medical history, neurological examination, neuropsychological assessment, fluorodeoxyglucose-PET, high-resolution 3T-MRI, long-term scalp-EEG, and invasive SEEG recordings. All patients had invasive SEEG recordings obtained by implanting multiple depth electrodes, each containing 10-18 contacts (2 mm long) separated by 1.5- or 5-mm contact spacing. The SEEG recordings were performed as part of routine clinical management, in line with French national guidelines [3]. Recordings were stored separately for each seizure, with seizure onset and offset times marked by expert epileptologists. For stimulation-induced seizures, stimulation parameters (channels, frequency, amplitude, pulse width and duration) were additionally provided. SEEG recordings during rest were stored separately. Following electrode implantation, a cranial CT-scan was performed to obtain locations of electrodes in the brain.

**Virtual epileptic cohort data.** The virtual epileptic cohort of 30 patients is provided in a BIDS-iEEG compatible format [17]. Following this format, each patient's synthetic data are saved in two categories: simulated data and derived data. The simulated data contain simulated SEEG time series. The derived data contains structural information extracted from brain imaging scans (T1-MRI, DW-MRI and CT-scan) and underlying model parameters used to generate the simulated data.

For each patient, synthetic SEEG time series are provided in BrainVision format (**ieeg** folder in S1 Fig). These synthetic time series are grouped into three different folders for each type (*ses-01*: simulated seizure, *ses-02*: stimulated seizure, *ses-03*: stimulated interictal spikes). Each synthetic SEEG file contains the EZ hypothesis type used in its filename (*VEPhypothesis* or *ClinicalHypothesis*). Also, if multiple synthetic SEEG files are provided, they each have a unique run number (starting from run-01). For all synthetic SEEG files, electrode names and coordinates are provided as *tsv* files. For every patient, the number of simulated brain activities for each condition are summarized in Table 1.

In addition, we provided structural information (**struct** folder in S1 Fig), notably their connectome and gain matrix. The connectome comes in the form of a ZIP file (TVB-compatible data format) and contains information about connectivity weights (MxM matrix, M = 162 brain regions), connectivity centers, center orientations, connectivity areas and volumes, tract lengths and cortical/non-cortical region flags (for more information, see [59]). The gain matrix is saved as a *tsv* file, this MxN matrix contains M regions and N sensors and maps the simulated brain activity from the brain region level to the SEEG sensors, thus obtaining synthetic SEEG time series.

Finally, corresponding model and simulator parameters are provided for each synthetic SEEG file (**parameters** folder in S1 Fig). Stimulation parameters are also provided for stimulation-induced synthetic seizures. The synthetic time series on the brain source level are also provided for each synthetic SEEG file. A VEP atlas [18] is provided as a tsv file for mapping source labels to brain region names.

## Data processing

The data processing method used here has been described in [13]. Here, we briefly explain the method used. To construct the virtual epileptic patients, we first preprocessed the T1-MRI and DW-MRI data. Volumetric segmentation and cortical surface reconstruction were obtained from the patient-specific T1-MRI data using the recon-all pipeline of the FreeSurfer software package. The cortical surface was parcellated according to the VEP atlas [18]. We used the MRtrix software package to process the DW-MRI, employing an iterative algorithm to estimate the response functions and subsequently used constrained spherical deconvolution to derive the fiber orientation distribution functions. The iFOD2 algorithm was used

to sample 15 million tracts. The structural connectome was constructed by assigning and counting the streamlines to and from each VEP brain region. This results in a $162x162$ connectivity matrix which is symmetric (there is no directionality information available in the white matter fibers). The diagonal entries of the connectome matrix were set to 0 to exclude self-connections within areas and the matrix was normalized so that the maximum value was equal to one. We obtained the location of the SEEG contacts from post-implantation CT scans using GARDEL as part of the EpiTools software package [60]. Then we coregistered the contact positions from the CT scan space to the T1-MRI scan space of each patient.

## Neural mass models

A neural mass model describes the activity of a population of neurons, thus it can describe the local activity of a brain region. It is defined by a set of differential equations that govern their dynamics. Just like brain regions are connected through long range white fibers, neural masses are linked through the structural connectome to form a whole-brain network. The global equation for such a model can be given by

$$\dot{\psi}_i(t) = F(\psi_i(t)) + K\sum_{j=1}^{L} W(i,j)S(\psi_i(t),\psi_j(t)) \tag{1}$$

where $\psi_i(t)$ is a state vector of neural activity at brain region $i$ and time $t$. $\dot{\psi}$ is the temporal derivative of the state vector. $F$ is a function of the state and captures the local neural activity. In our case, $F$ reflects the Epileptor model, described above. $W$ is a matrix of heterogeneous connection strengths between node $i$ and $j$. $S$ is a coupling function of the local state $\psi_i$ and the distant delayed state $\psi_j$. That a node receives input through the network is given by the sum across the number of nodes $L$ and scaled by a constant $K$. In this paper, this set of differential equations is solved using an Euler integration scheme with a step size of 0.5 ms.

**Forward solution with neural mass models.** Mapping the neural activity from the sources (VEP brain regions) to the sensors (SEEG contacts) is done by solving the forward problem and estimating a source-to-sensor matrix (gain matrix). As sources for our model, we used the vertices of the pial surface and volume bounding surfaces for the cortical and subcortical regions respectively. Surfaces are represented as triangular meshes. We estimate that the matrix $g_{j,k}$ from source brain region $j$ to sensor $k$ is equal to the sum of the inverse of the squared Euclidean distance $d_{i,k}$ from vertex $i$ to sensor $k$ weighted by the area $a_i$ of the vertex on the surface.

$$g_{j,k} = \sum_{i=0}^{N_j} \frac{a_i}{d_{i,k}^2} \tag{2}$$

Here vertex $i$ belongs to region $j$ which has $N_j$ vertices in total. The area $a_i$ of vertex $i$ is obtained by summing up one-third of the area of all the neighboring triangles. Vertices belonging to the same brain region are summed to obtain the gain for a single region of our brain network model. The resulting gain matrix has dimensions $MxN$, with $M$ being the number of regions and $N$ the number of sensors. Matrix multiplication of the simulated source activity with the gain matrix yields the simulated SEEG signals.

## The epileptor model

We used whole-brain network models to generate synthetic SEEG time series. Within a brain network model, each brain region is represented as a node and the connections between

regions are represented as edges. The brain regions are obtained by the FreeSurfer parcellation using the VEP atlas. The connection strength between regions is inferred from the structural connectome derived from DW-MRI data. The brain activity of each brain region is represented by a neural mass model, here we used the phenomenological 6D Epileptor model. There are 6 coupled differential equations in this model, which model 3 neural populations acting on a fast, intermediate and slow time scale.

$$\dot{x}_1 = y_1 - f_1(x_1, x_2) - z + I_{ext1}$$
$$\dot{y}_1 = c - dx_1^2 - y_1$$
$$\dot{x}_2 = -y_2 + x_2 - x_2^3 + I_{ext2} + 0.002g - 0.3(z - 3.5)$$
$$\dot{y}_2 = \frac{1}{\tau}(-y_2 + f_2(x_2))$$
$$\dot{z} = r\left(4(x_1 - x_0) - z + f_3(z) + K\sum_{j=1}^{N} C_{i,j}(x_1^j - x_1^i)\right)$$
$$\dot{g} = -0.01(g - 0.1x_1)$$

$$\text{(3)}$$

where

$$f_1(x_1, x_2) = \begin{cases} ax_1^3 - bx_1^2 & \text{if } x_1 < 0 \\ -(m - x_2 + 0.6(z - 4)^2)x_1 & \text{if } x_1 \geq 0 \end{cases}$$

$$f_2(x_2) = \begin{cases} 0 & \text{if } x_2 < -0.25 \\ a_2(x_2 + 0.25) & \text{if } x_2 \geq -0.25 \end{cases}$$

$$f_3(z) = \begin{cases} -0.1z^7 & \text{if } z < 0 \\ 0 & \text{if } z \geq 0 \end{cases}$$

The state variables $x_1$ and $y_1$ describe the activity of the neural population acting on a fast time scale to model fast discharges during epileptic seizures. The state variables $x_2$ and $y_2$ describe the activity of the neural population acting on an intermediate time scale to to model spike and wave phenomena during seizures. The state variable z acts on a slow time scale and drives the system autonomously in and out of the ictal state. In addition, the state variable $g$ acts as a low-pass filter of the coupling from $x_1$ to $x_2$ and generates the preictal and ictal spikes.

The excitability parameter $x_0$ represents the degree of epileptogenicity and determines whether the system converges towards an ictal or healthy state. If $x_0 > x_0c$, where $x_0c$ is the critical value of epileptogenicity, the Epileptor shows seizure activity autonomously and is referred to as epileptogenic; otherwise the Epileptor is in its (healthy) equilibrium state and does not trigger seizures autonomously. The default parameters are $r = 0.00035$, $\tau = 10$, $I_{ext1} = 3.1$, $I_{ext2} = 0.45$, $a = 1$, $a_2 = 6$, $b = 3$, $c = 1$, $d = 5$ and $m = 0$.

In addition, the Epileptor model is coupled to N other Epileptors via a linar approximation of permittivity coupling $K\sum_{j=1}^{N} C_{i,j}(x_1^j - x_1^i)$. In this coupling term, $K$ scales the global connectivity and can be varied between simulations to investigate different scenarios. The patient's connectome is represented by $C_{i,j}$ which defines region-to-region connection weights.

### The epileptor-stimulation model

To model stimulated seizures, we needed to determine the relationship between stimulus and brain activity. During stimulated seizures, we observed a slow increase in oscillatory response,

followed by a sudden switch to the seizure state, likely due to ion imbalances (e.g., extracellular potassium) reaching a critical threshold. To model this, We used the phenomenological Epileptor model. We transformed the parameter $m$ into a variable that accumulates stimulus effects slowly which influences the excitability of the model. When reaching a critical seizure threshold value $m_{thresh}$, it can push the system from its normal state to the seizure state via the permittivity variable $z$, which guides the system in and out of seizures (see S3 Fig). The extended Epileptor-stimulation model is as follows:

$$\dot{x}_1 = y_1 - f_1(x_1, x_2) - z + I_{ext1} + nI_{stim}$$

$$\dot{y}_1 = c - dx_1^2 - y_1$$

$$\dot{x}_2 = -y_2 + x_2 - x_2^3 + I_{ext2} + 0.002g - 0.3(z - 3.5)$$

$$\dot{y}_2 = \frac{1}{\tau}(-y_2 + f_2(x_2))$$

$$\dot{z} = r\left(4(x_1 - x_0 - H(m - m_{thresh})) - z + f_3(z) + K\sum_{j=1}^{N} C_{i,j}(x_1^j - x_1^i)\right)$$

$$\dot{g} = -0.01(g - 0.1x_1)$$

$$\dot{m} = r_2(k|I_{stim}| - 0.3m) \tag{4}$$

where

$$f_1(x_1, x_2) = \begin{cases} ax_1^3 - bx_1^2 & \text{if } x_1 < 0 \\ -(m - x_2 + 0.6(z - 4)^2)x_1 & \text{if } x_1 \geq 0 \end{cases}$$

$$f_2(x_2) = \begin{cases} 0 & \text{if } x_2 < -0.25 \\ a_2(x_2 + 0.25) & \text{if } x_2 \geq -0.25 \end{cases}$$

$$f_3(z) = \begin{cases} -0.1z^7 & \text{if } z < 0 \\ 0 & \text{if } z \geq 0 \end{cases}$$

All default parameters are the same as in the original Epileptor model, except for the additional parameters: $m_{thresh} = 1.5$, $k = 20$, $r = 0.00035$, $r_2 = 0.006$, $n = 3$, $x_0 = -2.2$. The Epileptor-stimulation model is coupled to N other Epileptors-stimulation using the same permittivity coupling described in the previous section. $I_{stim}$ is a time varying input describing the perturbation signal at each time step, and matches the clinically applied stimulus waveform. Spatially, it is weighted by a scalar corresponding to the estimated electric field magnitude for each brain region (for more detail, see the subsection below). $H$ is the Heaviside function, $m_{thresh}$ is the threshold for $m$ which when crossed changes the state of the system by pushing it in the upstate.

### Calculation of the electric field of SEEG stimulation

The French guidelines on SEEG stimulation state that bipolar and biphasic current should be used between two contiguous contacts to target a region of interest [3]. In this setting, one contact acts as a cathode (negative electric potential, sink of current) and the other one as an anode (positive electric potential, source of current). Current flows from the anode to the cathode, hyperpolarizing the neural elements nearest the anode and depolarizing the neural elements nearest the anode. This generates a local electric field in the area where the electrodes are located. A bipolar configuration is preferred for SEEG stimulation because it may

be less likely to elicit side effects thanks to the current being more focused than a monopolar configuration and less likely to spread into adjacent structures [61,62]. A symmetrical biphasic pulse waveform is used to reduce tissue damage by producing a zero net-charge. The parameters used clinically are restricted to frequencies of either 1 Hz or 50 Hz, weak amplitudes ranging from 0.5 to 5 mA, pulse widths of 500–3000 microseconds and durations of 0.5–40 seconds (short duration for 50 Hz stimulation and longer duration for 1 Hz stimulation).

We modeled the stimulus that was clinically applied by generating a bipolar signal following the stimulation parameters. The electric field generated by the stimulus was estimated by approximating the electrode contacts as point sources ($q_+$ and $q_-$), which is sufficiently accurate for our neural mass modeling approach [57]. We then mapped the stimulus signal onto the parcellated brain areas based on the distance, resulting in an estimated electric field at the whole-brain level. We used the estimated field magnitude $|\vec{E}(\vec{r})|$ as an input to the $I_{stim}$ parameter of the Epileptor-stimulation model (4). The field magnitude at a brain location $\vec{r}$ and permittivity $\epsilon$ is computed as:

$$|\vec{E}(\vec{r})| = \left| \frac{q}{4\pi\epsilon} \left( \frac{\vec{r} - \vec{r_{q+}}}{|\vec{r} - \vec{r_{q+}}|^3} - \frac{\vec{r} - \vec{r_{q-}}}{|\vec{r} - \vec{r_{q-}}|^3} \right) \right| \tag{5}$$

where $\epsilon$ is the permittivity of the medium. We set $\epsilon = 1$, as the electric field magnitude input to the phenomenological model is scaled based on the range (minimum and maximum) of the model variables.

## Spontaneous seizures

To simulate brain dynamics for each patient, we used their virtual brain model and the extended Epileptor model, parametrized by the patient's EZ hypothesis. The EZ hypothesis was based on the VEP pipeline as a first approach and on the clinical hypothesis as a second approach.

To obtain spontaneous seizures, two main parameters were adjusted in the Epileptor model: $x_0$ and $K$. Firstly, the $x_0$ parameter determines regional excitability. To simulate spontaneous seizures, the epileptogenicity heatmap obtained from the EZ hypothesis was translated into $x_0$ parameter values. For this, normalized epileptogenicity values were linearly transformed into an $x_0$ range of [–2.2,–1.2], such that seizures occurred autonomously. For $x_0$ values below –2.062 the model settles into a fixed point in the down-state which corresponds to an interictal state. For $-2.062 < x_0 < -1.025$ the model generates a stable oscillation in the up-state which corresponds to a seizure-like event (SLE). For $x_0$ values above –1.025 the model settles into a stable fixed point in the up-state. The brain regions that are epileptogenic have $x_0$ values corresponding to the SLE state and brain regions that are non-epileptogenic have values corresponding to the interictal state.

Secondly, the global coupling parameter (noted as $K$) is a key parameter which influences the resulting seizure dynamics. This parameter adjusts the coupling strength between nodes, which are connected to each other via a fast-to-slow coupling, also known as permittivity coupling [51]. This parameter is adjusted according to the empirical SEEG recordings. Both $x_0$ and $K$ determine the simulated spatiotemporal seizure dynamics. For instance, seizures can propagate to non-epileptogenic areas if they're connected to epileptogenic areas. It can also happen than seizures do not propagate to an epileptogenic area in particular cases when that area is connected to multiple healthy regions, which act as seizure inhibitors.

## Stimulated seizures

To generate stimulated seizures, we adjusted the parameters $x_0$, $m_{thresh}$ and $I_{stim}$. The parameter $K$ value was the same as in the spontaneous seizure. We used the same epileptogenicity heat map from the EZ hypothesis of the spontaneous seizures. To trigger seizures by external stimulation rather then them occuring spontaneously, we set the excitability $x_0$ values to a sub-critical threshold for seizing, linearly mapping them to an $x_0$ range of $[-2.2,-2.07]$. The seizure threshold $m_{thresh}$ parameter was also set from the EZ hypothesis, by linearly mapping them to a $[0.5,10]$ range. Epileptogenic brain regions have lower seizure thresholds than healthy brain regions.

Bipolar current stimulation via SEEG electrodes is applied in order to trigger seizures for epileptogenic zone diagnosis. For this, clinicians stimulate across multiple electrode pairs and across stimulation parameters. This is not done systematically, it rather follows the clinician's hypothesis of the epileptogenic zone and their experience with stimulation parameters. We have selected the clinical stimulation parameters which induced a seizure in the patient. We used the same stimulation electrodes and stimulation parameters in our model. The generated effects of the stimulus are mapped onto the brain regions using the sensor-to-source forward solution. This resulted in an estimated electric field magnitude, represented as scalar weights across brain regions, with the strongest weights located near the stimulating electrode pair. The $I_{stim}$ parameter was defined for each brain region depending on the stimulus weights. It then varied in time following the stimulus bipolar waveform. The variable $m$ is related to regional excitability and it depends on this parameter. When $I_{stim}$ is non-zero, $m$ slowly increases, otherwise it slowly returns to baseline. When $m > m_{thresh}$, the system is pushed to the seizure state. Then, the structural connectivity influences the spatio-temporal triggered seizure dynamics.

**Varying stimulation location.** We wanted to interrogate the robustness of our model by systematically changing stimulation location and comparing the outcome to the empirical stimulation-induced seizure. For this we followed the following steps for each of seven patients which had stimulation-induced seizures in the clinic.

We used the same brain models for each patient that were used for the virtual epileptic cohort. We reproduced the same parameters using the VEP hypothesis, taken from the cohort's **derivatives** database, with the only free parameter being stimulation location. To define this parameter, we grouped all SEEG contacts into four categories using the empirical stimulation location as a reference point. If we define $d_e$ as the distance in cm from the empirical stimulation location, Dist1: $d_e \leq 1$; Dist2: $d_e \in [1,2]$; Dist3: $d_e \in [2,3]$; and Dist4: $d_e \geq 3$. Next, we randomly stimulated up to 10 pairs of electrodes within the four main distance groups located outside of the empirical location. This resulted in up to 40 simulations containing the modeled stimulation input and whole-brain response (induced seizure or no seizure).

The comparative metrics between the synthetic SEEG stimulation responses and the empirical SEEG stimulation-induced seizure were performed (Fig 5). In total, 243 simulations were generated and compared against the empirical SEEG recordings.

**Varying stimulation amplitude.** We wanted to interrogate the robustness of our model by systematically changing stimulation amplitude and comparing the outcome to the empirical stimulation-induced seizure. For this we followed the following steps for each of seven patients which had stimulation-induced seizures in the clinic.

We used the same brain models for each patient that were used for the virtual epileptic cohort. We reproduced the same parameters using the VEP hypothesis, taken from the cohort's **derivatives** database, with the only free parameter being stimulation amplitude.

In the empirical stimulation parameters, all patients had a stimulation amplitude between 1.8 mA and 2.2 mA (mean = 2 mA, std = 0.12). We varied this parameter at the following amplitudes: 0.5 mA, 1 mA, 3 mA and 4 mA. We ran simulations for each stimulation amplitude, resulting in four simulations per patient containing the modeled stimulation input and whole brain response (induced seizure or no seizure).

The comparative metrics between the synthetic SEEG stimulation responses and the empirical SEEG stimulation-induced seizure were performed (Fig 5). In total, 28 simulations were generated and compared against the empirical SEEG recordings.

## Interictal spikes

We generated interictal activity for each patient, containing normal activity with interictal spiking in certain locations. We adjusted the parameters $I_{ext}$ and $x_0$. We used the same global coupling parameter $K$ of the spontaneous seizures.

We set $I_{ext} = 6.0$ and mapped the epileptogenicity values to an $x_0$ range of $[-3, -2.8]$, with additive stochastic noise for irregular spiking. This allowed for interictal spikes to be obtained by the model, but it is not the only method [36]. The combination of $x_0$ values close to the seizure threshold alongside the structural connectivity scaled by $K$ yields the interictal zone network for each patient. This results in interictal spike time series which are personalized to each patient.

The present literature relates the epileptogenic zone network to the interictal spike network [42]. Thus, we used the epileptogenicity heat map obtained from both the VEP hypothesis and the clinical hypothesis to set the EZN close to the critical threshold for seizure-like events.

**Interictal spike detection.**  A spike is defined as a transient distinguished from background activity, with pointed peak and duration between 20–70 ms and varying amplitude typically $> 50 \mu V$ [63]. A bandpass butterworth noncasual filter was applied on the data (lowcut 1 Hz, highcut 70 Hz). To detect a spike, we looked for the peaks of the signal which crossed a defined threshold. The threshold was defined following the spike detection method in [19]. If we define $|x|$ as the bandpass-filtered signal, then the threshold is equal to $4\sigma$, where $\sigma = median\{\frac{|x|}{0.6745}\}$. In this case, $\sigma$ is an estimate of the standard deviation of the background noise. The standard deviation of the signal could lead to very high threshold values, especially in cases with high firing rates and large spike amplitudes. By using the estimation based on the median, the interference of the spikes is diminished (see a demo in [19]). In the empirical spike case, we only took spikes that had an amplitude above $25 \mu V$. In addition, if two or more peaks were detected within 250 ms (or 256 timesteps in our algorithm) they were counted as a single spike, to ensure that poly-spikes are not counted as multiple spikes.

**Estimation of interictal spike count.**  To evaluate the synthetic interictal time series, we compared their interictal spike count (IIS) against the empirical recording. We used 15 minutes of SEEG interictal activity and 10000 timesteps of synthetic SEEG interictal time series, with a step size 0.05.

If the total number of spikes for a channel $i$ is $S_i$ and there are $N$ channels, then the IIS for that channel was computed as follows,

$$IIS_i = \frac{S_i}{\sum_{i=1}^{N} S_i} \tag{6}$$

As a result, we obtained two vectors of length $N$, containing the interictal spike count for the synthetic and the empirical SEEG time series. We compared these two vectors using the Pearson correlation coefficient.

## Randomized cohort

We generated three randomized cohorts for each simulation type: spontaneous seizures, stimulation induced seizures and interictal spikes. Each randomized cohort contained in total 15 synthetic SEEG time series for each EZ hypothesis (VEP hypothesis or Clinical hypothesis). The following approach was applied to generate one randomized cohort.

First, for each patient, the same parameters used to run the virtual epileptic cohort simulations were reused with the exception of the parameter $x_0$. For this parameter, instead of using the patient's own EZ hypothesis, we select it randomly from another virtual epileptic cohort patient of the cohort. This operation was performed 3 times for each patient. Thus, 3 synthetic simulations are obtained for each patient using a random EZ hypothesis.

Next, we compared the synthetic SEEG seizure from the randomized cohort with the empirical SEEG recording of that patient. If the patient has multiple SEEG recordings for the same type, we select one and we compare all 3 simulations to this SEEG simulation. We do this because seizure features and interictal features within the same patient tend to be more similar than those between patients.

We followed this procedure for 5 patients of our cohort, resulting in 15 simulated seizures for each EZ hypothesis. We used the randomized cohort simulations to compute similarity metrics between the empirical and simulated data. We then compared the same metrics between the virtual epileptic cohort and the randomized cohort. The virtual epileptic cohort cohort showed a significantly higher resemblance with the empirical seizure features as compared to the randomized cohort. This shows that patient-specific EZ hypothesis plays an important role for simulating spatio-temporal seizure dynamics.

## Comparing simulated and empirical SEEG traces

To compare the simulated SEEG time series to the empirical SEEG recording we first captured spatio-temporal features in both data. First, we computed an envelope function for each SEEG electrode as explained in [13] (see also S6 Fig). We used this envelope to mark each electrode as either seizure (containing seizure activity) or non-seizure (not containing any seizure activity) electrode. If the envelope's peak amplitude crossed a determined threshold, the SEEG electrodes were marked as seizure electrodes, otherwise they were marked as non-seizure electrodes. The threshold was the same for all electrodes and it was manually determined to be higher than the envelope's baseline amplitude . We compared the overlap of seizure and non-seizure electrodes between the simulated and empirical SEEG. In the electrodes where seizure activity was marked, we used the timepoints when the envelope jumped from its baseline and returned back to baseline to mark seizure onset and offset, respectively. Using this information, we binarized all SEEG traces in time: 0 for no seizure, 1 for seizure (Fig 3B). We then compared the binarized simulated and empirical SEEG using 2D pearson correlation.

Next, we divided SEEG electrodes marked as seizure electrodes into two groups: seizure onset (SO) and seizure propagation (SP) electrodes (Fig 3C). A seizure electrode was marked as SO when its onset time was belonged to the first few seconds of the total seizure length (corresponding to the first 5–15% of the total seizure duration), otherwise it was marked as SP. Then, the Jaccard similarity coefficient was employed to compare synthetic and empirical SO groups, and synthetic and empirical SP groups.

We computed these measurements for each pair of simulated SEEG and it's corresponding empirical SEEG, for both the spontaneous and stimulation-induced seizures. We repeated the same measurements for both the Virtual Epileptic Cohort (VEC) and the Randomized Cohort (RC). All measurements were plotted together and a boxplot was overlayed to compare the

VEC against the RC (Figs 3D and 5A). The boxplot is constructed as follows. Middle box represents the interquartile range (IQR), with a line at the median. The whiskers extend from the box to the data point lying within 1.5× the IQR. Points past the whiskers are marked as fliers.

We compared how significantly higher the mean for each metric of VEC was to the mean for each corresponding metric of RC using permutation testing [64] ($H_0$: mean(VEC) $\leq$ mean(RC), $H_1$: mean(VEC) > mean(RC)). For all measurements, p<0.001 therefore it is very unlikely that VEC performs better than RC by chance.

The complete set of all measurements that were performed can be found in S8 Fig and S1 File.

**Computing the 2D pearson correlation.** We compare the two binarized images using Pearson correlation and overlap.

$$r = \frac{\sum (x_i - \bar{x})(y_i - \bar{y})}{\sqrt{\sum (x_i - \bar{x})^2 \sum (y_i - \bar{y})^2}} \tag{7}$$

where $x_i$ is the binary value of a pixel in the empirical case, $y_i$ is the binary value of a pixel in the simulated case. Note: $r \in [-1,1]$.

**Binary overlap.**

$$Binary_{overlap} = \frac{|E_{bin} \cap S_{bin}|}{|E_{bin}|} \tag{8}$$

The ratio between the amount of identical timepoints divided by the entire amount of timepoints.

**Jaccard similarity coefficient.** We compare the seizure onset (SO) and seizure propagation (SP) groups using Jaccard similarity coefficient.

$$SO_{Jaccard} = \frac{|E_{SO} \cap S_{SO}|}{|E_{SO} \cup E_{SO}|} \quad \text{and} \quad SP_{Jaccard} = \frac{|E_{SP} \cap S_{SP}|}{|E_{SP} \cup E_{SP}|} \tag{9}$$

where $E_{SO}$, $E_{SP}$ are the empirical seizure onset and seizure propagation channels, respectively. $S_{SO}$, $S_{SP}$ are the simulated seizure onset and seizure propagation channels, respectively.

**Signal power.** For each SEEG bipolar sensor, the signal power is computed for both the empirical and the synthetic time series and plotted in 3D (e.g., Fig 2, right panel).

$$P = \frac{1}{N} \sum_{t=0}^{N} s_t^2 \tag{10}$$

where $s_t$ represents the electrode's signal amplitude at timepoint $t$, $N$ is the total number of time points, $P$ is the signal power for one electrode. The signal power across all electrodes is then normalized between 0 and 1.

## Permutation test

We performed a permutation test [64] for each metric ($H_0 : mean(VEC) \leq mean(RC)$, $H_1 : mean(VEC) > mean(RC)$). First, a test statistic was chosen, here it was the difference in means between the VEC group and the RC group. Next, all metric values from the VEC group and the RC group were pooled together into a single dataset. Then, the data were randomly shuffled 200,000 times. At each shuffling iteration, the pooled dataset was divided back into two groups of the same sizes as the VEC and RC groups. The difference in means between the

two groups was performed at each iteration. Finally, the test statistic computed before the permutations was compared to the distribution of 200,000 permuted statistics, by calculating a p-value as the proportion of permuted statistics that are as extreme or more extreme than the observed statistic. For us, the p-value was the probability of obtaining permuted differences in means higher than the one between the VEC and RC groups. If this probability is low (i.e. less than 0.01), it means that there is a significant difference between the VEC and RC group.

## Supporting information

**S1 Fig. BIDS-IEEG structure of the virtual epileptic cohort.** The structure contains the simulated SEEG spontaneous seizures, stimulated seizures and interictal spikes in the main folder of each patient (named sub-001, sub-002, etc.). The parameters used to obtain the simulated time series are detailed in the derivatives folder for each of the EZ hypothesis (either VEP hypothesis or clinical hypothesis).
(PDF)

**S2 Fig. Two different seizure types from the same patient.** (**A**) Focal seizure occurring in left temporal lobe. (**B**) Propagated seizure with onset in left temporal lobe and propagation to contra lateral hemisphere. For both cases, simulated and empirical SEEG seizures are shown. Red vertical lines indicate seizure onset and seizure offset. Signal power for all SEEG channels is shown in 3D in axial and sagittal plane.
(PNG)

**S3 Fig. Example time series for Epileptor-stimulation model.** Above, empirical SEEG recording with stimulation artefact present. Below, simulated time series using the Epileptor-stimulation model. Variables of the model $x_1$, $z$, $m$ and $I_{stim}$ are shown. (**A**) Example with 1 Hz, 1 mA stimulation and no seizure being induced. (**B**) Example with 1 Hz, 1.5 mA stimulation and a seizure is induced. Asterisks indicate the time series containing seizure activity.
(PNG)

**S4 Fig. Simulation examples with different stimulation amplitude from virtualized patient 10.** Double arrow indicates stimulation period. Seven channels are plotted in bipolar montage (out of 116 total bipolar channels). Vertical red lines indicate seizure onset and seizure offset. For all plots (**B**), (**C**) and (**D**), upper plots show simulated time series at the SEEG level with stimulation applied at different amplitudes. Lower plots show variables $x_1 - x_2$ in red, $z$ in green and $m$ in purple evolving for the same simulation for the region left hippocampus anterior. (A) Empirical SEEG recording plot of a stimulation-induced seizure. Stimulation was applied at amplitude 1 mA using channels TB'1 [+] and TB'2 [-], at frequency 50 Hz, pulse width 1 ms and duration 4 s. Reconstructed SEEG electrodes are shown on the left and stimulation location is plotted in red. (B) Upper plot, synthetic SEEG time series of simulated brain activity with stimulation applied at 0.5 mA amplitude. All other stimulation parameters are identical to the empirical parameters. Here, a seizure is not induced after the stimulation is applied. Lower plot, the same simulated activity for the left hippocampus anterior, showing the variable $m$ did not cross the seizure threshold, defined at 3.5 and corresponding variables staying in the normal state. (C) Synthetic SEEG time series of the stimulation-induced seizure at 1 mA amplitude. Here, the same stimulation parameters as the ones applied empirically were used. Following the stimulation, a seizure is induced in the left hippocampus anterior, propagating later on to connected brain structures. Lower plot showing the variable $m$ crossed the seizure threshold and the system is kicked to the seizure state. (D) Synthetic SEEG time series plot of simulated brain activity with stimulation applied at

3 mA amplitude. Following the stimulation, a seizure is induced in the left hippocampus anterior and propagating later on to connected brain structures. Lower plot showing the variable *m* crossed the seizure threshold and the system is kicked to the seizure state.
(PNG)

**S5 Fig. Simulation examples with different stimulation location from virtualized patient 10.** The stimulation location was chosen by randomly selecting a pair of electrodes within a certain radius from the empirical stimulation location. Double arrow indicates stimulation period. Seven channels are plotted in bipolar montage (out of 116 total bipolar channels). Vertical red lines indicate seizure onset and seizure offset. (**A**) Upper plot, empirical SEEG recording of a stimulation-induced seizure. Stimulation was applied at amplitude 1 mA using channels TB'1 [+] and TB'2 [-], at frequency 50 Hz, pulse width 1 ms and duration 4 s. Lower plot, corresponding simulated time series of a stimulation-induced seizure. (**B**) Simulated time series of stimulation applied by electrodes located within 1 cm distance from the empirical stimulation location (TB'1-2). (**C**) Simulated time series of stimulation applied by electrodes located between 1 and 2 cm distance from the empirical stimulation location. (**D**) Simulated time series of stimulation applied by electrodes located between 2 and 3 cm distance from the empirical stimulation location. (**E**) Simulated time series of stimulation applied by electrodes located more than 3 cm away from the empirical stimulation location. As the stimulus is applied increasingly further away from the empirical stimulation location, the seizure dynamics progressively changes from the empirical post-stimulation response.
(PNG)

**S6 Fig. Comparing simulated and empirical SEEG by extracting data features from the SEEG time series: spontaneous seizure example.** (**A**) Timeseries plot of a few SEEG channels (right: simulated SEEG time series, left: empirical SEEG time series). Overlayed in red and black are the envelope data features for each SEEG channel, indicating seizure and non seizure channels respectively. Green points indicate estimated seizure onset times. (**B**) Envelope data features overlayed for all SEEG channels (left: simulated, right:empirical). Horizontal red lines indicate chosen threshold to categorize each channel as either seizure (above threshold) or non-seizure channel (below threshold). (**C**) Binary plot of the same SEEG channels, where black indicates seizure activity and white indicates no seizure activity.
(PNG)

**S7 Fig. Similarity metrics between simulated and empirical SEEG time series using the clinical hypothesis for the EZ ground truth.** Comparison of simulated SEEG signals with empirical recordings for the virtual epileptic cohort (VEC, in blue) and the randomized cohort (RC, in red). The clinical hypothesis was used to inform the excitability parameters in the model. (**A**) Boxplot of four main metrics comparing spontaneous seizures against synthetic seizures. Red dots indicate mean values. (**B**) Boxplot of four main metrics comparing synthetic against empirical stimulation-induced seizures. (**C**) Boxplot of interictalspike (IIS) count correlation metric. $^{****}p-value < 0.0001$, $^{***}p-value < 0.001$, $^{**}p-value < 0.01$; permutation test.
(PNG)

**S8 Fig. Overall statistics for the VEC dataset compared to the RC dataset for spontaneous seizures.** To compare the simulated SEEG time series against the empirical SEEG, sixteen metrics were tested in total. In blue, mean and standard deviation for the VEC cohort. In red, mean and standard deviation for the randomized cohort.
(PNG)

**S9 Fig. Overall statistics for the VEC dataset compared to the RC dataset for stimulated seizures.** To compare the simulated SEEG time series against the empirical SEEG, sixteen metrics were tested in total. In blue, mean and standard deviation for the VEC cohort. In red, mean and standard deviation for the randomized cohort.
(PNG)

**S10 Fig. Grouped metrics following surgical outcome.** Following their Engel score, patients were grouped in either the seizure-free group (Engel score I) or the not-seizure-free group (Engel scores II, III and IV). First row shows metrics from synthetic data using the VEP hypothesis. The second row uses the clinical hypothesis. In each plot, metrics from seizure-free patients are plotted on the left side, whearese metrics from not-seizure-free patients are plotted on the right side. A boxplot is overlaid over all individual data points and a violin plot from the same data points is shown on the right next to it.
(PNG)

**S1 File.** The file contains a description of the complete set of metrics used to compare simulated and empirical SEEG seizure time series.
(PDF)

## Acknowledgments

We express our sincere appreciation to Samuel Medina Villalon and Dr. Romain Carron for their assistance with the empirical retrospective dataset.

## Author contributions

**Conceptualization:** Borana Dollomaja, Huifang E. Wang, Viktor K. Jirsa.

**Data curation:** Borana Dollomaja, Huifang E. Wang, Maxime Guye, Julia Makhalova, Fabrice Bartolomei.

**Formal analysis:** Borana Dollomaja.

**Funding acquisition:** Huifang E. Wang.

**Investigation:** Borana Dollomaja.

**Methodology:** Borana Dollomaja, Huifang E. Wang, Viktor K. Jirsa.

**Project administration:** Huifang E. Wang.

**Resources:** Julia Makhalova, Fabrice Bartolomei.

**Software:** Borana Dollomaja.

**Supervision:** Huifang E. Wang, Viktor K. Jirsa.

**Validation:** Borana Dollomaja.

**Visualization:** Borana Dollomaja.

**Writing – original draft:** Borana Dollomaja, Huifang E. Wang.

**Writing – review & editing:** Borana Dollomaja, Huifang E. Wang, Maxime Guye, Julia Makhalova, Fabrice Bartolomei, Viktor K. Jirsa.

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
