## [Decision Letter · Decision Letter 0]

27 Nov 2024

PCOMPBIOL-D-24-01804Virtual epilepsy patient cohort: generation and evaluationPLOS Computational BiologyDear Dr. Dollomaja,

Thank you for submitting your manuscript to PLOS Computational Biology. After careful consideration, we feel that it has merit but does not fully meet PLOS Computational Biology's publication criteria as it currently stands. Therefore, we invite you to submit a revised version of the manuscript that addresses the points raised during the review process.

Please submit your revised manuscript within 60 days Jan 27 2025 11:59PM. If you will need more time than this to complete your revisions, please reply to this message or contact the journal office at ploscompbiol@plos.org. Please include the following items when submitting your revised manuscript: * A rebuttal letter that responds to each point raised by the editor and reviewer(s). You should upload this letter as a separate file labeled 'Response to Reviewers'. This file does not need to include responses to formatting updates and technical items listed in the 'Journal Requirements' section below. * A marked-up copy of your manuscript that highlights changes made to the original version. You should upload this as a separate file labeled 'Revised Manuscript with Track Changes'. * An unmarked version of your revised paper without tracked changes. You should upload this as a separate file labeled 'Manuscript'. If you would like to make changes to your financial disclosure, competing interests statement, or data availability statement, please make these updates within the submission form at the time of resubmission. Guidelines for resubmitting your figure files are available below the reviewer comments at the end of this letter. We look forward to receiving your revised manuscript. Kind regards, Peter Neal TaylorAcademic EditorPLOS Computational Biology  Andrea E. MartinSection EditorPLOS Computational Biology  Feilim Mac GabhannEditor-in-ChiefPLOS Computational BiologyJason PapinEditor-in-ChiefPLOS Computational Biology  **Journal Requirements:**

1) Please ensure that the CRediT author contributions listed for every co-author are completed accurately and in full. At this stage, the following Authors/Authors require contributions: Borana Dollomaja, Huifang E. Wang, Maxime Guye, Julia Makhalova, Fabrice Bartolomei, and Viktor Jirsa. Please ensure that the full contributions of each author are acknowledged in the "Add/Edit/Remove Authors" section of our submission form. The list of CRediT author contributions may be found here: https://journals.plos.org/ploscompbiol/s/authorship#loc-author-contributions 

Please respond directly to this email and provide any known details concerning your material's license terms and permissions required for reuse, even if you have not yet obtained copyright permissions or are unsure of your material's copyright compatibility. Once you have responded and addressed all other outstanding technical requirements, you may resubmit your manuscript within Editorial Manager.  Potential Copyright Issues: - Figures: 1, 2, 4, 5, 6, S2, S4, S5, and Graphical Abstract.. Please confirm whether you drew the images / clip-art within the figure panels by hand. If you did not draw the images, please provide a link to the source of the images or icons and their license / terms of use; or written permission from the copyright holder to publish the images or icons under our CC BY 4.0 license. Alternatively, you may replace the images with open source alternatives. See these open source resources you may use to replace images / clip-art: - https://commons.wikimedia.org - https://openclipart.org/.**Reviewers' comments:**

Reviewer's Responses to Questions

**Comments to the Authors:**

Reviewer #1: The authors present a synthetic dataset of 30 virtual epileptic patients. This is potentially useful to the community studying seizures and resective surgery. The manuscript is well presented, providing a thorough description of the methods used to generate and analyse the data.

The aim of the manuscript is clearly to present the dataset and illustrate capabilities, rather than to present new results. The presentation of interictal spikes and stimulus evoked seizures is novel only within the VEP framework, but again presents a potentially useful resource for investigating the links between seizures and stimulus evoked dynamics. I would encourage the authors to expand on the points below to improve the presentation of the context of the work.

The VEP framework is not a universal ground truth for the generation of seizures nor the evaluation of epilepsy surgery. It would be of benefit to explore this more thoroughly in the manuscript. For example, though the assumptions are implicit in the description of the model, it would be beneficial to end users to spell these out upfront, in the main part of the manuscript, interpret them biologically and critique them. For example, in this model, simulated seizures are due to a build up in "z", which is designed to push nodes through bifurcations into seizures (in what sense are they spontaneous? Is recurrent a better description?). This is only one of a few possible models of recurrent or spontaneous seizure generation, none of which have been shown to be universally true (see work by Lopes da Silva, Piotr Suffczynski, Gerold Baier etc.. ). Similarly, in this model, propagation patterns emerge due to the distribution of the excitability parameter, and the seizure onset zone is equated to the epileptogenic zone. This may or may not be the case, see Lopes et al. 2019 (https://www.frontiersin.org/journals/computational-neuroscience/articles/10.3389/fncom.2019.00025/full) for a computational evaluation of the role of network structure and the interplay with excitability. These are important points to highlight since the synthetic data will be useful for exploring methods to detect the EZ only for cases in which these assumptions hold, and readers should be appropriately informed.

Figure 1 shows the first example of stimulated seizures. It would be important to clarify in what sense these are 'seizures', rather than, for example, afterdischarges. For example in Gollwitzer et al. 2018 ( https://pubmed.ncbi.nlm.nih.gov/30269940/ ) the term afterdischarges is used, with seizures emerging from these in only 3.5% of cases. Ley et al. 2022 (

https://www.sciencedirect.com/science/article/pii/S1388245722001699

) provides some useful background, and considers stimulation evoked activity to be 'seizures' if they are the same as a patient's seizures electrographically and have the same semiology. Furthermore they note that afterdischarges are not necessarily a reliable marker of the seizure onset zone (or epileptogenic zone), which implies that the mechanisms (underlying network and excitability patterns) producing these and seizures may not be the same, i.e. may not be reproducible with the same underlying structural connectivity or distributed excitability in the model. Again, this should be clarified for the reader who is interested in using the presented synthetic data to understand stimulation evoked activity.

The data consists of seizures, evoked 'seizures' and spikes. This is motivated in the introduction since these types of activity can be used to constrain the EZ. The relationship between the EZ, spikes and evoked activity is not straightforward though and background dynamics are also being shown to contain information useful for delineating the EZ [e.g. Taylor et al. 2022 https://academic.oup.com/brain/article/145/3/939/6514463 ].

"Here we present for the first time synthetic seizures triggered by SEEG stimulation" in introduction and "Here, we present for the first time…" in discussion are a bit of a stretch I think. Many previous models have presented stimulus evoked seizures (see e.g. Baier et al. 2017 https://www.frontiersin.org/journals/computational-neuroscience/articles/10.3389/fncom.2017.00025/full and wider literature).

The section 2.1 could be improved by adding further (brief) details on the comparisons that were made, for example a brief account of what it means to reproduce the clinically defined EZ with a precision of 0.6.

In section 2.4.1 it would be useful to clarify to what extent the stimulation parameters are the same as those used clinically. The model is not biophysical so the real electrical disturbance cannot be recreated.

The end of the paragraph in discussion starting "We simulated.." contains a section on neural field models that only refers to the authors' previous work. I would suggest to widen this to include work from other members of the community.

Finally, although it is potentially useful to present synthetic data, the community would benefit hugely from sharing of the actual recordings. There are precedents for this - e.g. databases with surgery records and EEG from Mayo and MGH (at https://www.ieeg.org/) also

see Wong et al. 2023 (https://pmc.ncbi.nlm.nih.gov/articles/PMC10235576/ ). It would be interesting to hear why the authors have chosen not to share the original EEG, if indeed that is the case.

Reviewer #2: The Ms describes a dataset of synthetic seizures generated in 30 virtual epilepsy

patients, using patient-specific anatomical and functional connectivity data and the epileptor model to simulate seizure dynamics. The dataset is publicly available, and the manuscript argues that this synthetic cohort can serve as a valuable resource for evaluating data analysis methodologies. The primary advantage of synthetic seizures is the availability of ground truth regarding the epileptogenic zone (EZ), which is often unavailable in empirical data. Overall, I have no major concerns with the manuscript, but I provide several comments below that should be addressed before it is ready for acceptance. Addressing these points will mainly enhance the clarity and presentation of the manuscript.

1. Figure 1 Workflow Consistency: In Figure 1, the simulated seizures are presented in the upper panel, while empirical seizures appear below. In contrast, Figures 2, 4, and 6 reverse this order, showing empirical signals above simulated ones. Is there a specific reason for this reversal? Does it reflect a difference between the workflow for generating the virtual cohort and the synthetic signals depicted in Figures 2, 4, and 6?

2. EZ Hypothesis Clarification: Figure 1’s caption states that the EZ hypothesis, derived from ictal SEEG data, informs the excitability parameter for each brain region. The study uses two EZ hypotheses: one based on the Virtual Epileptic Patient (VEP) hypothesis from spontaneous seizures, and another clinically defined by experts. Please clarify which hypothesis is used in the Figure 1 workflow and whether this aligns with the depicted sequence (simulated seizures first, empirical seizures second).

3. Seizure Propagation Metrics: The manuscript mentions that the performance for the virtual epileptic cohort is significantly better than the randomized cohort (p < 0.001), as shown in Figure 5A. However, please also note that Seizure Propagation metrics showed no differences between the two cohorts.

4. Boxplot Presentation Issues:

4.1 Some boxplots depicting cohort statistics appear odd and need improvement. For example, in Figure 3D, the VEC boxplot for “Seizure Propagation” shows most points below the median except for two. Similarly, in Figure 5A for the RC boxplots, the middle box contains only two points, though it should include 50% of the data (approximately 7 or 8 points). This may be due to overlapping points. If it so, I suggest to overlay box plots with swarm plots.

4.2 The RC cohort points have a very light color shade, making them hard to discern. Consider using a stronger shade for better visibility.

4.3 The dark red point in each boxplot, representing the mean value, is described as “dark red” but appears as just a darker shade of the RC cohort points. This is misleading, as it also represents the mean for the VEC. Using a different, unrelated color (e.g., black or violet) would help avoid confusion.

5. Redundant Paragraph: The paragraph in lines 88–98 includes information that overlaps with earlier descriptions in lines 79–87, such as the use of synthetic data for validating analysis methods and the role of the virtual brain twin. Please revise to reduce redundancy.

6. Acronym Clarification: The acronym VEC is introduced in line 152 but only explained in the Figure 3 legend. Please define it upon first use.

7. Parameter Explanation: The meaning of the parameters k and q in Equation (5) is not explained. Additionally, it appears that k in Equation (5) differs from k in Equation (4). Please clarify.

**Have the authors made all data and (if applicable) computational code underlying the findings in their manuscript fully available?**

Reviewer #1: **No: **it seems EEG data is not provided

Reviewer #2: Yes

PLOS authors have the option to publish the peer review history of their article (what does this mean?). If published, this will include your full peer review and any attached files.

Reviewer #1: No

Reviewer #2: No

**Figure resubmission:** While revising your submission, please upload your figure files to the Preflight Analysis and Conversion Engine (PACE) digital diagnostic tool, https://pacev2.apexcovantage.com/. PACE helps ensure that figures meet PLOS requirements. To use PACE, you must first register as a user. Registration is free. Then, login and navigate to the UPLOAD tab, where you will find detailed instructions on how to use the tool. If you encounter any issues or have any questions when using PACE, please email PLOS at figures@plos.org. Please note that Supporting Information files do not need this step. If there are other versions of figure files still present in your submission file inventory at resubmission, please replace them with the PACE-processed versions.
---

## [Decision Letter · Decision Letter 1]

25 Feb 2025

Dear Miss Dollomaja,

We are pleased to inform you that your manuscript 'Virtual epilepsy patient cohort: generation and evaluation' has been provisionally accepted for publication in PLOS Computational Biology.

Best regards,

Peter Neal Taylor

Academic Editor

PLOS Computational Biology

Andrea E. Martin

Section Editor

PLOS Computational Biology

Reviewer's Responses to Questions

**Comments to the Authors:**

Reviewer #1: The authors have addressed my comments sufficiently, thank you. Regarding data sharing, thank you for providing a detailed response. I would maintain that the simulated data is no substitute for the original data and at some point as a community we must overcome these barriers to progress our understanding of neurological disorders and improve treatment. The authors note that sharing data is challenging. It seems that it is not impossible and hopefully they can in future obtain resources to undertake the case-by-case procedure referred to.

**Have the authors made all data and (if applicable) computational code underlying the findings in their manuscript fully available?**

Reviewer #1: None

PLOS authors have the option to publish the peer review history of their article (what does this mean?). If published, this will include your full peer review and any attached files.

Reviewer #1: No

---

## [Editor Report · Acceptance letter]

PCOMPBIOL-D-24-01804R1

Virtual epilepsy patient cohort: Generation and evaluation

Dear Dr Dollomaja,

I am pleased to inform you that your manuscript has been formally accepted for publication in PLOS Computational Biology. Your manuscript is now with our production department and you will be notified of the publication date in due course.

With kind regards,

Anita Estes
